# Solving Most Systems of Random Quadratic Equations

**Gang Wang**[⋆,∗]    **Georgios B. Giannakis**[∗]    **Yousef Saad**[†]    **Jie Chen**[⋆]

[⋆]Key Lab of Intell. Contr. and Decision of Complex Syst., Beijing Inst. of Technology
[∗]Digital Tech. Center & Dept. of Electrical and Computer Eng., Univ. of Minnesota
[†]Department of Computer Science and Engineering, Univ. of Minnesota
{gangwang, georgios, saad}@umn.edu; chenjie@bit.edu.cn.

## Abstract

This paper deals with finding an $n$-dimensional solution $\boldsymbol{x}$ to a system of quadratic equations $y_i = |\langle \boldsymbol{a}_i, \boldsymbol{x} \rangle|^2$, $1 \leq i \leq m$, which in general is known to be NP-hard. We put forth a novel procedure, that starts with a *weighted maximal correlation initialization* obtainable with a few power iterations, followed by successive refinements based on *iteratively reweighted gradient-type iterations*. The novel techniques distinguish themselves from prior works by the inclusion of a fresh (re)weighting regularization. For certain random measurement models, the proposed procedure returns the true solution $\boldsymbol{x}$ with high probability in time proportional to reading the data $\{(\boldsymbol{a}_i; y_i)\}_{1 \leq i \leq m}$, provided that the number $m$ of equations is some constant $c > 0$ times the number $n$ of unknowns, that is, $m \geq cn$. Empirically, the upshots of this contribution are: i) perfect signal recovery in the high-dimensional regime given only an *information-theoretic limit number* of equations; and, ii) (near-)optimal statistical accuracy in the presence of additive noise. Extensive numerical tests using both synthetic data and real images corroborate its improved signal recovery performance and computational efficiency relative to state-of-the-art approaches.

## 1 Introduction

One is often faced with solving quadratic equations of the form $y_i = |\langle \boldsymbol{a}_i, \boldsymbol{x} \rangle|^2$, or equivalently,

$$\psi_i = |\langle \boldsymbol{a}_i, \boldsymbol{x} \rangle|, \quad 1 \leq i \leq m \tag{1}$$

where $\boldsymbol{x} \in \mathbb{R}^n / \mathbb{C}^n$ (hereafter, symbol "$A/B$" denotes either $A$ or $B$) is the wanted unknown $n \times 1$ vector, while given observations $\psi_i$ and feature vectors $\boldsymbol{a}_i \in \mathbb{R}^n / \mathbb{C}^n$ that are collectively stacked in the data vector $\boldsymbol{\psi} := [\psi_i]_{1 \leq i \leq m}$ and the $m \times n$ sensing matrix $\boldsymbol{A} := [\boldsymbol{a}_i]_{1 \leq i \leq m}$, respectively. Put differently, given information about the (squared) modulus of the inner products of the signal vector $\boldsymbol{x}$ and several known design vectors $\boldsymbol{a}_i$, can one reconstruct exactly (up to a global phase factor) $\boldsymbol{x}$, or alternatively, the missing phase of $\langle \boldsymbol{a}_i, \boldsymbol{x} \rangle$? In fact, much effort has been devoted to determining the number of such equations necessary and/or sufficient for the uniqueness of the solution $\boldsymbol{x}$; see e.g., [1, 8]. It has been proved that $m \geq 2n - 1$ ($m \geq 4n - 4$) generic [1] (which includes the case of random vectors) real (complex) vectors $\boldsymbol{a}_i$ are sufficient for uniquely determining an $n$-dimensional real (complex) vector $\boldsymbol{x}$ [1, Theorem 2.8], [8], while in the real case $m = 2n - 1$ is shown also necessary [1]. In this sense, the number $m = 2n - 1$ of equations as in (1) can be regarded as the information-theoretic limit for such a quadratic system to be uniquely solvable.

In diverse physical sciences and engineering fields, it is impossible or very difficult to record phase measurements. The problem of recovering the signal or phase from magnitude measurements only, also commonly known as phase retrieval, emerges naturally [10, 11]. Relevant application domains include e.g., X-ray crystallography, astronomy, microscopy, ptychography, and coherent diffraction imaging [21]. In such setups, optical measurement and detection systems record solely the photon flux, which is proportional to the (squared) magnitude of the field, but not the phase. Problem (1) in its squared form, on the other hand, can be readily recast as an instance of nonconvex quadratically constrained quadratic programming, that subsumes as special cases several well-known combinatorial optimization problems involving Boolean variables, e.g., the NP-complete stone problem [2, Sec. 3.4.1]. A related task of this kind is that of estimating the mixture of linear regressions, where the latent membership indicators can be converted into the missing phases [29]. Although of simple form and practical relevance across different fields, solving systems of nonlinear equations is arguably the most difficult problem in all of the numerical computations [19, Page 355].

*Notation*: Lower- (upper-) case boldface letters denote vectors (matrices), e.g., $\boldsymbol{a} \in \mathbb{R}^n$ ($\boldsymbol{A} \in \mathbb{R}^{m \times n}$). Calligraphic letters are reserved for sets. The floor operation $\lfloor c \rfloor$ gives the largest integer no greater than the given real quantity $c > 0$, the cardinality $|\mathcal{S}|$ counts the number of elements in set $\mathcal{S}$, and $\|\boldsymbol{x}\|$ denotes the Euclidean norm of $\boldsymbol{x}$. Since for any phase $\phi \in \mathbb{R}$, vectors $\boldsymbol{x} \in \mathbb{C}^n$ and $e^{j\phi}\boldsymbol{x}$ are indistinguishable given $\{\psi_i\}$ in (1), let $\mathrm{dist}(\boldsymbol{z}, \boldsymbol{x}) := \min_{\phi \in [0, 2\pi)} \|\boldsymbol{z} - \boldsymbol{x}e^{j\phi}\|$ be the Euclidean distance of any estimate $\boldsymbol{z} \in \mathbb{C}^n$ to the solution set $\{e^{j\phi}\boldsymbol{x}\}_{0 \le \phi < 2\pi}$ of (1); in particular, $\phi = 0/\pi$ in the real case.

## 1.1 Prior contributions

Following the least-squares (LS) criterion (which coincides with the maximum likelihood (ML) one assuming additive white Gaussian noise), the problem of solving quadratic equations can be naturally recast as an empirical loss minimization

$$\underset{\boldsymbol{z} \in \mathbb{R}^n / \mathbb{C}^n}{\text{minimize}} \quad L(\boldsymbol{z}) := \frac{1}{m} \sum_{i=1}^{m} \ell(\boldsymbol{z}; \psi_i/y_i) \tag{2}$$

where one can choose to work with the *amplitude-based* loss $\ell(\boldsymbol{z}; \psi_i) := (\psi_i - |\langle \boldsymbol{a}_i, \boldsymbol{z} \rangle|)^2/2$ [28, 30], or the *intensity-based* one $\ell(\boldsymbol{z}; y_i) := (y_i - |\langle \boldsymbol{a}_i, \boldsymbol{z} \rangle|^2)^2/2$ [3], and its related Poisson likelihood $\ell(\boldsymbol{z}; y_i) := y_i \log(|\langle \boldsymbol{a}_i, \boldsymbol{z} \rangle|^2) - |\langle \boldsymbol{a}_i, \boldsymbol{z} \rangle|^2$ [7]. Either way, the objective functional $L(\boldsymbol{z})$ is nonconvex; hence, it is generally NP-hard and computationally intractable to compute the ML or LS estimate.

Minimizing the squared modulus-based LS loss in (2), several numerical polynomial-time algorithms have been devised via convex programming for certain choices of design vectors $\boldsymbol{a}_i$ [4, 25]. Such convex paradigms first rely on the matrix-lifting technique to express all squared modulus terms into linear ones in a new rank-1 matrix variable, followed by solving a convex semi-definite program (SDP) after dropping the rank constraint. It has been established that perfect recovery and (near-)optimal statistical accuracy are achieved in noiseless and noisy settings respectively with an optimal-order number of measurements [4]. In terms of computational efficiency however, such lifting-based convex approaches entail storing and solving for an $n \times n$ semi-definite matrix from $m$ general SDP constraints, whose computational complexity in the worst case scales as $n^{4.5} \log 1/\epsilon$ for $m \approx n$ [25], which is not scalable. Another recent line of convex relaxation [12], [13] reformulated the problem of phase retrieval as that of sparse signal recovery, and solved a linear program in the natural parameter vector domain. Although exact signal recovery can be established assuming an accurate enough anchor vector, its empirical performance is in general not competitive with state-of-the-art phase retrieval approaches.

Recent proposals advocate suitably initialized iterative procedures for coping with certain nonconvex formulations directly; see e.g., algorithms abbreviated as AltMinPhase, (R/P)WF, (M)TWF, (S)TAF [16, 3, 7, 26, 28, 27, 30, 22, 6, 24], as well as a prox-linear algorithm [9]. These nonconvex approaches operate directly upon vector optimization variables, thus leading to significant computational advantages over their convex counterparts. With random features, they can be interpreted as performing stochastic optimization over acquired examples $\{(\boldsymbol{a}_i; \psi_i/y_i)\}_{1 \le i \le m}$ to approximately minimize the population risk functional $\bar{L}(\boldsymbol{z}) := \mathbb{E}_{(\boldsymbol{a}_i, \psi_i/y_i)}[\ell(\boldsymbol{z}; \psi_i/y_i)]$. It is well documented that minimizing nonconvex functionals is generally intractable due to existence of multiple critical points [17]. Assuming Gaussian sensing vectors however, such nonconvex paradigms can provably locate the global optimum, several of which also achieve optimal (statistical) guarantees. Specifically,

starting with a judiciously designed initial guess, successive improvement is effected by means of a sequence of (truncated) (generalized) gradient-type iterations given by

$$z^{t+1} := z^t - \frac{\mu^t}{m} \sum_{i \in \mathcal{T}^{t+1}} \nabla \ell_i(z^t; \psi_i/y_i), \quad t = 0, 1, \ldots \tag{3}$$

where $z^t$ denotes the estimate returned by the algorithm at the $t$-th iteration, $\mu^t > 0$ is learning rate that can be pre-selected or found via e.g., the backtracking line search strategy, and $\nabla \ell(z^t, \psi_i/y_i)$ represents the (generalized) gradient of the modulus- or squared modulus-based LS loss evaluated at $z^t$. Here, $\mathcal{T}^{t+1}$ denotes some time-varying index set signifying the per-iteration gradient truncation.

Although they achieve optimal statistical guarantees in both noiseless and noisy settings, state-of-the-art (convex and nonconvex) approaches studied under Gaussian designs, empirically require stable recovery of a number of equations (several) times larger than the information-theoretic limit [7, 3, 30]. As a matter of fact, when there are numerously enough measurements (on the order of $n$ up to some polylog factors), the squared modulus-based LS functional admits benign geometric structure in the sense that [23]: i) all local minimizers are also global; and, ii) there always exists a negative directional curvature at every saddle point. In a nutshell, the grand challenge of tackling systems of random quadratic equations remains to develop algorithms capable of achieving perfect recovery and statistical accuracy when the number of measurements approaches the information limit.

## 1.2    This work

Building upon but going beyond the scope of the aforementioned nonconvex paradigms, the present paper puts forward a novel iterative linear-time scheme, namely, time proportional to that required by the processor to scan all the data $\{(a_i; \psi_i)\}_{1 \le i \le m}$, that we term *reweighted amplitude flow*, and henceforth, abbreviate as RAF. Our methodology is capable of solving noiseless random quadratic equations exactly, yielding an estimate of (near)-optimal statistical accuracy from noisy modulus observations. Exactness and accuracy hold with high probability and without extra assumption on the unknown signal vector $x$, provided that the ratio $m/n$ of the number of equations to that of the unknowns is larger than a certain constant. Empirically, our approach is shown able to ensure exact recovery of high-dimensional unstructured signals given a *minimal* number of equations, where $m/n$ in the real case can be as small as 2. The new twist here is to leverage judiciously designed yet conceptually simple (re)weighting regularization techniques to enhance existing initializations and also gradient refinements. An informal depiction of our RAF methodology is given in two stages as follows, with rigorous details deferred to Section 3:

**S1) Weighted maximal correlation initialization:** Obtain an initializer $z^0$ maximally correlated with a carefully selected subset $\mathcal{S} \subsetneq \mathcal{M} := \{1, 2, \ldots, m\}$ of feature vectors $a_i$, whose contributions toward constructing $z^0$ are judiciously weighted by suitable parameters $\{w_i^0 > 0\}_{i \in \mathcal{S}}$.

**S2) Iteratively reweighted "gradient-like" iterations:** Loop over $0 \le t \le T$:

$$z^{t+1} = z^t - \frac{\mu^t}{m} \sum_{i=1}^m w_i^t \nabla \ell(z^t; \psi_i) \tag{4}$$

for some time-varying weighting parameters $\{w_i^t \ge 0\}$, each possibly relying on the current iterate $z_t$ and the datum $(a_i; \psi_i)$.

Two attributes of the novel approach are worth highlighting next. First, albeit being a variant of the spectral initialization devised in [28], the initialization here [cf. S1)] is distinct in the sense that different importance is attached to each selected datum $(a_i; \psi_i)$. Likewise, the gradient flow [cf. S2)] weighs judiciously the search direction suggested by each datum $(a_i; \psi_i)$. In this manner, more robust initializations and more stable overall search directions can be constructed even based solely on a rather limited number of data samples. Moreover, with particular choices of the weights $w_i^t$'s (e.g., taking $0/1$ values), the developed methodology subsumes as special cases the recently proposed algorithms RWF [30] and TAF [28].

## 2    Algorithm: Reweighted Amplitude Flow

This section explains the intuition and basic principles behind each stage of the advocated RAF algorithm in detail. For analytical concreteness, we focus on the *real Gaussian model* with $x \in \mathbb{R}^n$,

and independent sensing vectors $\boldsymbol{a}_i \in \mathbb{R}^n \sim \mathcal{N}(\boldsymbol{0}, \boldsymbol{I})$ for all $1 \leq i \leq m$. Nonetheless, the presented approach can be directly applied when the complex Gaussian and the coded diffraction pattern (CDP) models are considered.

## 2.1 Weighted maximal correlation initialization

A key enabler of general nonconvex iterative heuristics' success in finding the global optimum is to seed them with an excellent starting point [14]. Indeed, several smart initialization strategies have been advocated for iterative phase retrieval algorithms; see e.g., the spectral initialization [16], [3] as well as its truncated variants [7], [28], [9], [30], [15]. One promising approach is the one pursued in [28], which is also shown robust to outliers in [9]. To hopefully approach the information-theoretic limit however, its performance may need further enhancement. Intuitively, it is increasingly challenging to improve the initialization (over state-of-the-art) as the number of acquired data samples approaches the information-theoretic limit.

In this context, we develop a more flexible initialization scheme based on the correlation property (as opposed to the orthogonality in [28]), in which the added benefit is the inclusion of a flexible weighting regularization technique to better balance the useful information exploited in the selected data. Similar to related approaches of the same kind, our strategy entails estimating both the norm $\|\boldsymbol{x}\|$ and the direction $\boldsymbol{x}/\|\boldsymbol{x}\|$ of $\boldsymbol{x}$. Leveraging the strong law of large numbers and the rotational invariance of Gaussian $\boldsymbol{a}_i$ vectors (the latter suffices to assume $\boldsymbol{x} = \|\boldsymbol{x}\|\boldsymbol{e}_1$, with $\boldsymbol{e}_1$ being the first canonical vector in $\mathbb{R}^n$), it is clear that

$$\frac{1}{m}\sum_{i=1}^{m}\psi_i^2 = \frac{1}{m}\sum_{i=1}^{m}\big|\langle \boldsymbol{a}_i, \|\boldsymbol{x}\|\boldsymbol{e}_1\rangle\big|^2 = \Big(\frac{1}{m}\sum_{i=1}^{m}a_{i,1}^2\Big)\|\boldsymbol{x}\|^2 \approx \|\boldsymbol{x}\|^2 \tag{5}$$

whereby $\|\boldsymbol{x}\|$ can be estimated to be $\sum_{i=1}^{m}\psi_i^2/m$. This estimate proves very accurate even with a limited number of data samples because $\sum_{i=1}^{m}a_{i,1}^2/m$ is unbiased and tightly concentrated.

The challenge thus lies in accurately estimating the direction of $\boldsymbol{x}$, or seeking a unit vector maximally aligned with $\boldsymbol{x}$. Toward this end, let us first present a variant of the initialization in [28]. Note that the larger the modulus $\psi_i$ of the inner-product between $\boldsymbol{a}_i$ and $\boldsymbol{x}$ is, the known design vector $\boldsymbol{a}_i$ is deemed *more correlated* to the unknown solution $\boldsymbol{x}$, hence bearing useful directional information of $\boldsymbol{x}$. Inspired by this fact and having available data $\{(\boldsymbol{a}_i; \psi_i)\}_{1\leq i\leq m}$, one can sort all (absolute) correlation coefficients $\{\psi_i\}_{1\leq i\leq m}$ in an ascending order, yielding ordered coefficients $0 < \psi_{[m]} \leq \cdots \leq \psi_{[2]} \leq \psi_{[1]}$. Sorting $m$ records takes time proportional to $\mathcal{O}(m\log m)$.[2] Let $\mathcal{S} \subsetneq \mathcal{M}$ denote the set of selected feature vectors $\boldsymbol{a}_i$ to be used for computing the initialization, which is to be designed next. Fix *a priori* the cardinality $|\mathcal{S}|$ to some integer on the order of $m$, say, $|\mathcal{S}| := \lfloor 3m/13 \rfloor$. It is then natural to *define* $\mathcal{S}$ to collect the $\boldsymbol{a}_i$ vectors that correspond to one of the largest $|\mathcal{S}|$ correlation coefficients $\{\psi_{[i]}\}_{1\leq i\leq|\mathcal{S}|}$, each of which can be thought of as pointing to (roughly) the direction of $\boldsymbol{x}$. Approximating the direction of $\boldsymbol{x}$ therefore boils down to finding a vector to maximize its correlation with the subset $\mathcal{S}$ of selected directional vectors $\boldsymbol{a}_i$. Succinctly, the wanted approximation vector can be efficiently found as the solution of

$$\underset{\|\boldsymbol{z}\|=1}{\text{maximize}} \quad \frac{1}{|\mathcal{S}|}\sum_{i\in\mathcal{S}}\big|\langle \boldsymbol{a}_i, \boldsymbol{z}\rangle\big|^2 = \boldsymbol{z}^*\Big(\frac{1}{|\mathcal{S}|}\sum_{i\in\mathcal{S}}\boldsymbol{a}_i\boldsymbol{a}_i^*\Big)\boldsymbol{z} \tag{6}$$

where the superscript $^*$ represents the transpose or the conjugate transpose that will be clear from the context. Upon scaling the unity-norm solution of (6) by the norm estimate obtained $\sum_{i=1}^{m}\psi_i^2/m$ in (5), to match the magnitude of $\boldsymbol{x}$, we will develop what we will henceforth refer to as maximal correlation initialization.

As long as $|\mathcal{S}|$ is chosen on the order of $m$, the maximal correlation method outperforms the spectral ones in [3, 16, 7], and has comparable performance to the orthogonality-promoting method [28]. Its performance around the information-limit however, is still not the best that we can hope for. Recall from (6) that all selected directional vectors $\{\boldsymbol{a}_i\}_{i\in\mathcal{S}}$ are treated *the same* in terms of their contributions to constructing the initialization. Nevertheless, according to our starting principle, this ordering information carried by the selected $\boldsymbol{a}_i$ vectors is not exploited by the initialization scheme in (6) and [28]. In other words, if for $i, j \in \mathcal{S}$, the correlation coefficient of $\psi_i$ with $\boldsymbol{a}_i$ is larger

than that of $\psi_j$ with $\boldsymbol{a}_j$, then $\boldsymbol{a}_i$ is deemed more correlated (with $\boldsymbol{x}$) than $\boldsymbol{a}_j$ is, hence bearing more useful information about the direction of $\boldsymbol{x}$. It is thus prudent to weigh more the selected $\boldsymbol{a}_i$ vectors associated with larger $\psi_i$ values. Given the ordering information $\psi_{[|\mathcal{S}|]} \leq \cdots \leq \psi_{[2]} \leq \psi_{[1]}$ available from the sorting procedure, a natural way to achieve this goal is weighting each $\boldsymbol{a}_i$ vector with simple monotonically increasing functions of $\psi_i$, say e.g., taking the weights $w_i^0 := \psi_i^\gamma$, $\forall i \in \mathcal{S}$ with the exponent parameter $\gamma \geq 0$ chosen to maintain the wanted ordering $w_{[|\mathcal{S}|]}^0 \leq \cdots \leq w_{[2]}^0 \leq w_{[1]}^0$. In a nutshell, a more flexible initialization strategy, that we refer to as *weighted maximal correlation*, can be summarized as follows

$$\tilde{\boldsymbol{z}}_0 := \arg\max_{\|\boldsymbol{z}\|=1} \ \boldsymbol{z}^*\Big(\frac{1}{|\mathcal{S}|}\sum_{i\in\mathcal{S}}\psi_i^\gamma \boldsymbol{a}_i \boldsymbol{a}_i^*\Big)\boldsymbol{z}. \tag{7}$$

For any given $\epsilon > 0$, the power method or the Lanczos algorithm can be called for to find an $\epsilon$-accurate solution to (7) in time proportional to $\mathcal{O}(n|\mathcal{S}|)$ [20], assuming a positive eigengap between the largest and the second largest eigenvalues of the matrix $(1/|\mathcal{S}|)\sum_{i\in\mathcal{S}}\psi_i^\gamma \boldsymbol{a}_i \boldsymbol{a}_i^*$, which is often true when $\{\boldsymbol{a}_i\}$ are sampled from continuous distribution. The proposed initialization can be obtained upon scaling $\tilde{\boldsymbol{z}}_0$ from (7) by the norm estimate in (5), to yield $\boldsymbol{z}_0 := (\sum_{i=1}^m \psi_i^2/m)\tilde{\boldsymbol{z}}_0$. By default, we take $\gamma := 1/2$ in all reported numerical implementations, yielding $w_i^0 := \sqrt{|\langle\boldsymbol{a}_i, \boldsymbol{x}\rangle|}$ for all $i \in \mathcal{S}$.

Regarding the initialization procedure in (7), we next highlight two features, whereas technical details and theoretical performance guarantees are provided in Section 3:

**F1)** The weights $\{w_i^0\}$ in the maximal correlation scheme enable leveraging useful information that each feature vector $\boldsymbol{a}_i$ may bear regarding the direction of $\boldsymbol{x}$.

**F2)** Taking $w_i^0 := \psi_i^\gamma$ for all $i \in \mathcal{S}$ and 0 otherwise, problem (7) can be equivalently rewritten as

$$\tilde{\boldsymbol{z}}_0 := \arg\max_{\|\boldsymbol{z}\|=1} \ \boldsymbol{z}^*\Big(\frac{1}{m}\sum_{i=1}^m w_i^0 \boldsymbol{a}_i \boldsymbol{a}_i^*\Big)\boldsymbol{z} \tag{8}$$

which subsumes previous initialization schemes with particular selections of weights $\{w_i^0\}$. For instance, the spectral initialization in [16, 3] is recovered by choosing $\mathcal{S} := \mathcal{M}$, and $w_i^0 := \psi_i^2$ for all $1 \leq i \leq m$.

For comparison, define

$$\text{Relative error} := \frac{\text{dist}(\boldsymbol{z}, \boldsymbol{x})}{\|\boldsymbol{x}\|}.$$

Throughout the paper, all simulated results were averaged over 100 Monte Carlo (MC) realizations, and each simulated scheme was implemented with their pertinent default parameters. Figure 1 evaluates the performance of the developed initialization relative to several state-of-the-art strategies, and also with the information limit number of data benchmarking the minimal number of samples required. It is clear that our initialization is: i) consistently better than the state-of-the-art; and, ii) stable as $n$ grows, which is in contrast to the instability encountered by the spectral ones [16, 3, 7, 30]. It is worth stressing that the more than 5% empirical advantage (relative to the best) at the challenging *information-theoretic benchmark* is nontrivial, and is one of the main RAF upshots. This advantage becomes increasingly pronounced as the ratio $m/n$ grows.

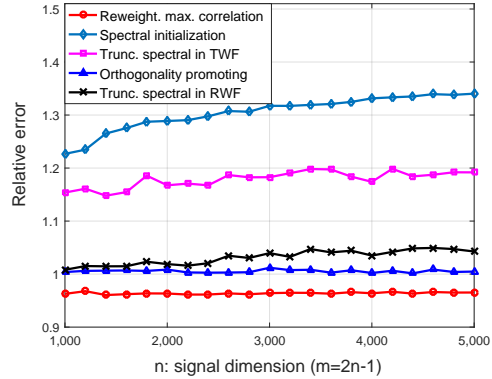

Figure 1: Relative initialization error for i.i.d. $\boldsymbol{a}_i \sim \mathcal{N}(\boldsymbol{0}, \boldsymbol{I}_{1,000})$, $1 \leq i \leq 1,999$.

## 2.2 Iteratively reweighted gradient flow

For independent data obeying the real Gaussian model, the direction that TAF moves along in stage S2) presented earlier is given by the following (generalized) gradient [28]:

$$\frac{1}{m}\sum_{i\in\mathcal{T}}\nabla\ell(\boldsymbol{z};\psi_i) = \frac{1}{m}\sum_{i\in\mathcal{T}}\Big(\boldsymbol{a}_i^*\boldsymbol{z} - \psi_i\frac{\boldsymbol{a}_i^*\boldsymbol{z}}{|\boldsymbol{a}_i^*\boldsymbol{z}|}\Big)\boldsymbol{a}_i \tag{9}$$

where the dependence on the iterate count $t$ is neglected for notational brevity, and the convention $a_i^* z / |a_i^* z| := 0$ is adopted when $a_i^* z = 0$.

Unfortunately, the (negative) gradient of the average in (9) generally may not point towards the true solution $x$ unless the current iterate $z$ is already very close to $x$. Therefore, moving along such a descent direction may not drag $z$ closer to $x$. To see this, consider an initial guess $z_0$ that has already been in a *basin of attraction* (i.e., a region within which there is only a unique stationary point) of $x$. Certainly, there are summands $(a_i^* z - \psi_i \frac{a_i^* z}{|a_i^* z|}) a_i$ in (9), that could give rise to "bad/misleading" gradient directions due to the erroneously estimated signs $\frac{a_i^* z}{|a_i^* z|} \neq \frac{a_i^* x}{|a_i^* x|}$ [28], or $(a_i^* z)(a_i^* x) < 0$ [30]. Those gradients as a whole may drag $z$ away from $x$, and hence out of the basin of attraction. Such an effect becomes increasingly severe as $m$ approaches the information-theoretic limit of $2n - 1$, thus rendering past approaches less effective in this case. Although this issue is somewhat remedied by TAF with a truncation procedure, its efficacy is limited due to misses of bad gradients and mis-rejections of meaningful ones around the information limit.

To address this challenge, reweighted amplitude flow effecting suitable gradient directions from *all* data samples $\{(a_i; \psi_i)\}_{1 \le i \le m}$ will be adopted in a (timely) adaptive fashion, namely introducing appropriate weights for all gradients to yield the update

$$z^{t+1} = z^t - \mu^t \nabla \ell_{\mathrm{rw}}(z^t; \psi_i), \quad t = 0, 1, \dots \quad (10)$$

The *reweighted gradient* $\nabla \ell_{\mathrm{rw}}(z^t)$ evaluated at the current point $z^t$ is given as

$$\nabla \ell_{\mathrm{rw}}(z) := \frac{1}{m} \sum_{i=1}^{m} w_i \nabla \ell(z; \psi_i) \quad (11)$$

for suitable weights $\{w_i\}_{1 \le i \le m}$ to be designed next.

To that end, we observe that the truncation criterion [28]

$$\mathcal{T} := \left\{ 1 \le i \le m : \frac{|a_i^* z|}{|a_i^* x|} \ge \alpha \right\} \quad (12)$$

with some given parameter $\alpha > 0$ suggests to include only gradient components associated with $|a_i^* z|$ of relatively large sizes. This is because gradients of sizable $|a_i^* z| / |a_i^* x|$ offer reliable and meaningful directions pointing to the truth $x$ with large probability [28]. As such, the ratio $|a_i^* z| / |a_i^* x|$ can be somewhat viewed as a confidence score about the reliability or meaningfulness of the corresponding gradient $\nabla \ell(z; \psi_i)$. Recognizing that confidence can vary, it is natural to distinguish the contributions that different gradients make to the overall search direction. An easy way is to attach large weights to the reliable gradients, and small weights to the spurious ones. Assume without loss of generality that $0 \le w_i \le 1$ for all $1 \le i \le m$; otherwise, lump the normalization factor achieving this into the learning rate $\mu^t$. Building upon this observation and leveraging the gradient reliability confidence score $|a_i^* z| / |a_i^* x|$, the weight per gradient $\nabla \ell(z; \psi_i)$ in RAF is designed to be

$$w_i := \frac{1}{1 + \beta_i / (|a_i^* z| / |a_i^* x|)}, \quad 1 \le i \le m \quad (13)$$

in which $\{\beta_i > 0\}_{1 \le i \le m}$ are some pre-selected parameters.

Regarding the proposed weighting criterion in (13), three remarks are in order, followed by the RAF algorithm summarized in Algorithm 1.

**R1)** The weights $\{w_i^t\}_{1 \le i \le m}$ are time adapted to $z^t$. One can also interpret the reweighted gradient flow $z^{t+1}$ in (10) as performing a single gradient step to minimize the *smooth reweighted* loss $\frac{1}{m} \sum_{i=1}^{m} w_i^t \ell(z; \psi_i)$ with starting point $z^t$; see also [4] for related ideas successfully exploited in the *iteratively reweighted least-squares* approach to compressive sampling.

**R2)** Note that the larger $|a_i^* z| / |a_i^* x|$ is, the larger $w_i$ will be. More importance will be attached to reliable gradients than to spurious ones. Gradients from *almost all* data points are are judiciously accounted for, which is in sharp contrast to [28], where withdrawn gradients do not contribute the information they carry.

**R3)** At the points $\{z\}$ where $a_i^* z = 0$ for certain $i \in \mathcal{M}$, the corresponding weight will be $w_i = 0$. That is, the losses $\ell(z; \psi_i)$ in (2) that are nonsmooth at points $z$ will be eliminated, to prevent their contribution to the reweighted gradient update in (10). Hence, the convergence analysis of RAF can be considerably simplified because it does not have to cope with the nonsmoothness of the objective function in (2).

## 2.3 Algorithmic parameters

To optimize the empirical performance and facilitate numerical implementations, choice of pertinent algorithmic parameters of RAF is independently discussed here. It is obvious that the RAF algorithm entails four parameters. Our theory and all experiments are based on: i) $|\mathcal{S}|/m \leq 0.25$; ii) $0 \leq \beta_i \leq 10$ for all $1 \leq i \leq m$; and, iii) $0 \leq \gamma \leq 1$. For convenience, a constant step size $\mu^t \equiv \mu > 0$ is suggested, but other step size rules such as backtracking line search with the reweighted objective work as well. As will be formalized in Section 3, RAF converges if the constant $\mu$ is not too large, with the upper bound depending in part on the selection of $\{\beta_i\}_{1 \leq i \leq m}$.

In the numerical tests presented in Sections 2 and 4, we take

$$|\mathcal{S}| := \lfloor 3m/13 \rfloor, \quad \beta_i \equiv \beta := 10, \quad \gamma := 0.5, \quad \text{and} \quad \mu := 2 \tag{14}$$

while larger step sizes $\mu > 0$ can be afforded for larger $m/n$ values.

---

**Algorithm 1** Reweighted Amplitude Flow

1: **Input:** Data $\{(\boldsymbol{a}_i; \psi_i)\}_{1 \leq i \leq m}$; maximum number of iterations $T$; step size $\mu^t = 2/6$ and weighting parameter $\beta_i = 10/5$ for real/complex Gaussian model; $|\mathcal{S}| = \lfloor 3m/13 \rfloor$, and $\gamma = 0.5$.
2: **Construct** $\mathcal{S}$ to include indices associated with the $|\mathcal{S}|$ largest entries among $\{\psi_i\}_{1 \leq i \leq m}$.
3: **Initialize** $\boldsymbol{z}^0 := \sqrt{\sum_{i=1}^{m} \psi_i^2/m}\, \tilde{\boldsymbol{z}}^0$ with $\tilde{\boldsymbol{z}}^0$ being the unit principal eigenvector of

$$\boldsymbol{Y} := \frac{1}{m} \sum_{i=1}^{m} w_i^0 \boldsymbol{a}_i \boldsymbol{a}_i^* \tag{15}$$

where $w_i^0 := \begin{cases} \psi_i^\gamma, & i \in \mathcal{S} \subseteq \mathcal{M} \\ 0, & \text{otherwise} \end{cases}$ for all $1 \leq i \leq m$.

4: **Loop: for** $t = 0$ **to** $T - 1$

$$\boldsymbol{z}^{t+1} = \boldsymbol{z}^t - \frac{\mu^t}{m} \sum_{i=1}^{m} w_i^t \left( \boldsymbol{a}_i^* \boldsymbol{z}^t - \psi_i \frac{\boldsymbol{a}_i^* \boldsymbol{z}^t}{|\boldsymbol{a}_i^* \boldsymbol{z}^t|} \right) \boldsymbol{a}_i \tag{16}$$

where $w_i^t := \frac{|\boldsymbol{a}_i^* \boldsymbol{z}^t|/\psi_i}{|\boldsymbol{a}_i^* \boldsymbol{z}^t|/\psi_i + \beta_i}$ for all $1 \leq i \leq m$.

5: **Output:** $\boldsymbol{z}^T$.

---

## 3  Main results

Our main results summarized in Theorem 1 next establish exact recovery under the real Gaussian model, whose proof is provided in the supplementary material. Our RAF approach however can be generalized readily to the complex Gaussian and CDP models.

**Theorem 1 (Exact recovery)** *Consider $m$ noiseless measurements $\boldsymbol{\psi} = |\boldsymbol{Ax}|$ for an arbitrary $\boldsymbol{x} \in \mathbb{R}^n$. If the data size $m \geq c_0|\mathcal{S}| \geq c_1 n$ and the step size $\mu \leq \mu_0$, then with probability at least $1 - c_3 \mathrm{e}^{-c_2 m}$, the reweighted amplitude flow's estimates $\boldsymbol{z}^t$ in Algorithm 1 obey*

$$\mathrm{dist}(\boldsymbol{z}^t, \boldsymbol{x}) \leq \frac{1}{10}(1 - \nu)^t \|\boldsymbol{x}\|, \quad t = 0, 1, \ldots \tag{17}$$

*where $c_0$, $c_1$, $c_2$, $c_3 > 0$, $0 < \nu < 1$, and $\mu_0 > 0$ are certain numerical constants depending on the choice of algorithmic parameters $|\mathcal{S}|$, $\beta$, $\gamma$, and $\mu$.*

According to Theorem 1, a few interesting properties of our RAF algorithm are worth highlighting. To start, RAF recovers the true solution exactly with high probability whenever the ratio $m/n$ of the number of equations to the unknowns exceeds some numerical constant. Expressed differently, RAF achieves the information-theoretic optimal order of sample complexity, which is consistent with the state-of-the-art including TWF [7], TAF [28], and RWF [30]. Notice that (17) also holds at $t = 0$, namely, $\mathrm{dist}(\boldsymbol{z}^0, \boldsymbol{x}) \leq \|\boldsymbol{x}\|/10$, therefore providing performance guarantees for the proposed initialization scheme (cf. Step 3 in Algorithm 1). Moreover, starting from this initial estimate, RAF converges linearly to the true solution $\boldsymbol{x}$. That is, to reach any $\epsilon$-relative solution accuracy (i.e., $\mathrm{dist}(\boldsymbol{z}^T, \boldsymbol{x}) \leq \epsilon\|\boldsymbol{x}\|$), it suffices to run at most $T = \mathcal{O}(\log 1/\epsilon)$ RAF iterations (cf. Step 4). This in

conjunction with the per-iteration complexity $\mathcal{O}(mn)$ confirms that RAF solves exactly a quadratic system in time $\mathcal{O}(mn \log {1}/{\epsilon})$, which is linear in $\mathcal{O}(mn)$, the time required to read the entire data $\{(\boldsymbol{a}_i; \psi_i)\}_{1 \le i \le m}$. Given the fact that the initialization stage can be performed in time $\mathcal{O}(n|\mathcal{S}|)$ and $|\mathcal{S}| < m$, the overall linear-time complexity of RAF is order-optimal.

Proof of Theorem 1 is provided in the supplementary material.

## 4 Simulated tests

Our theoretical findings about RAF have been corroborated with comprehensive numerical tests, a sample of which are discussed next. Performance of RAF is evaluated relative to the state-of-the-art (T)WF, RWF, and TAF in terms of the empirical success rate among 100 MC trials, where a success will be declared for a trial if the returned estimate incurs error

$$\frac{\left\| \boldsymbol{\psi} - |\boldsymbol{A}\boldsymbol{z}^T| \right\|}{\|\boldsymbol{x}\|} \le 10^{-5}$$

where the modulus operator $|\cdot|$ is understood element-wise. The real Gaussian model and the physically realizable CDPs were simulated in this section. For fairness, all schemes were implemented with their suggested parameter values. The true signal vector $\boldsymbol{x}$ was randomly generated using $\boldsymbol{x} \sim \mathcal{N}(\boldsymbol{0}, \boldsymbol{I})$, and the i.i.d. sensing vectors $\boldsymbol{a}_i$ $\boldsymbol{a}_i \sim \mathcal{N}(\boldsymbol{0}, \boldsymbol{I})$. Each scheme obtained the initial guess based on 200 power iterations, followed by

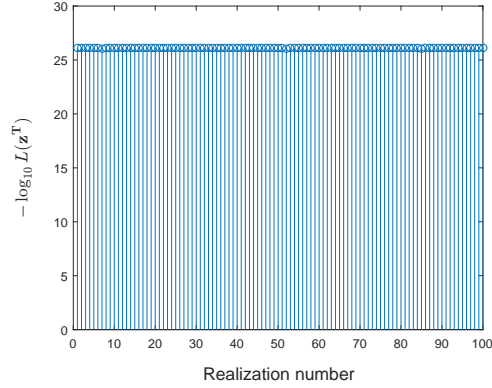

Figure 2: Function value $L(\boldsymbol{z}^T)$ by RAF for 100 MC realizations when $m = 2n - 1$.

a series of $T = 2,000$ (truncated/reweighted) gradient iterations. All experiments were performed using MATLAB on an Intel CPU @ 3.4 GHz (32 GB RAM) computer. For reproducibility, the Matlab code of the RAF algorithm is publicly available at `https://gangwg.github.io/RAF/`.

To demonstrate the power of RAF in the high-dimensional regime, the function value $L(\boldsymbol{z})$ in (2) evaluated at the returned estimate $\boldsymbol{z}^T$ for 100 independent trials is plotted (in negative logarithmic scale) in Fig. 2, where $m = 2n - 1 = 9,999$. It is self-evident that RAF succeeded in all trials even at this challenging information limit. To the best of our knowledge, RAF is the first algorithm that empirically recovers any solution exactly from a *minimal number* of random quadratic equations. Left panel in Fig. 3 further compares the empirical success rate of five schemes under the real Gaussian model with $n = 1,000$ and $m/n$ varying by 0.1 from 1 to 5. Evidently, the developed RAF achieves perfect recovery as soon as $m$ is about $2n$, where its competing alternatives do not work well. To demonstrate the stability and robustness of RAF in the presence of additive noise, the right panel in Fig. 3 depicts the normalized mean-square error

$$\text{NMSE} := \frac{\text{dist}^2(\boldsymbol{z}^T, \boldsymbol{x})}{\|\boldsymbol{x}\|^2}$$

as a function of the signal-to-noise ratio (SNR) for $m/n$ taking values $\{3, \ 4, \ 5\}$. The noise model

$$\psi_i = |\langle \boldsymbol{a}_i, \boldsymbol{x} \rangle| + \eta_i, \quad 1 \le i \le m$$

with $\boldsymbol{\eta} := [\eta_i]_{1 \le i \le m} \sim \mathcal{N}(\boldsymbol{0}, \sigma^2 \boldsymbol{I}_m)$ was employed, where $\sigma^2$ was set such that certain $\text{SNR} := 10 \log_{10}(\|\boldsymbol{A}\boldsymbol{x}\|^2 / m\sigma^2)$ values on the $x$-axis were achieved.

To examine the efficacy and scalability of RAF in real-world conditions, the last experiment entails the Galaxy image [3] depicted by a three-way array $\boldsymbol{X} \in \mathbb{R}^{1,080 \times 1,920 \times 3}$, whose first two coordinates encode the pixel locations, and the third the RGB color bands. Consider the physically realizable CDP model with random masks [3]. Letting $\boldsymbol{x} \in \mathbb{R}^n$ ($n \approx 2 \times 10^6$) be a vectorization of a certain band of $\boldsymbol{X}$, the CDP model with $K$ masks is

$$\boldsymbol{\psi}^{(k)} = |\boldsymbol{F}\boldsymbol{D}^{(k)}\boldsymbol{x}|, \quad 1 \le k \le K,$$

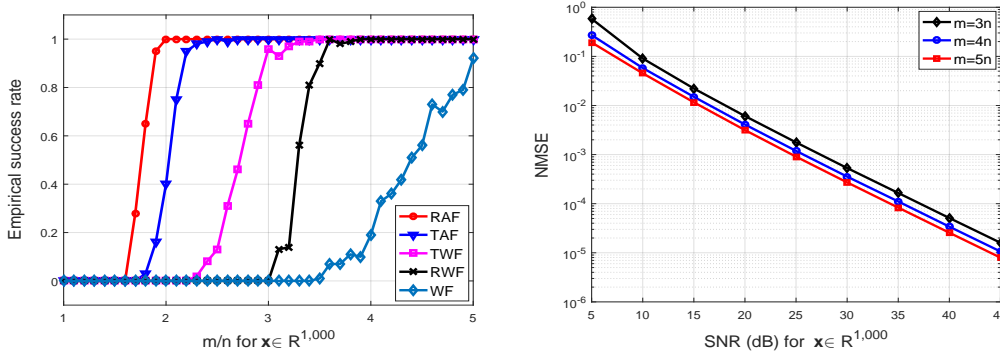

Figure 3: Real Gaussian model: Empirical success rate (Left); and, Relative MSE vs. SNR (Right).

where $\boldsymbol{F} \in \mathbb{C}^{n \times n}$ is a DFT matrix, and diagonal matrices $\boldsymbol{D}^{(k)}$ have their diagonal entries sampled uniformly at random from $\{1, -1, j, -j\}$ with $j := \sqrt{-1}$. Each $\boldsymbol{D}^{(k)}$ represents a random mask placed after the object to modulate the illumination patterns [5]. Implementing $K = 4$ masks, each algorithm performs independently over each band 100 power iterations for an initial guess, which was refined by 100 gradient iterations. Recovered images of TAF (left) and RAF (right) are displayed in Fig. 4, whose relative errors were $1.0347$ and $1.0715 \times 10^{-3}$, respectively. WF and TWF returned images of corresponding relative error $1.6870$ and $1.4211$, which are far away from the ground truth.

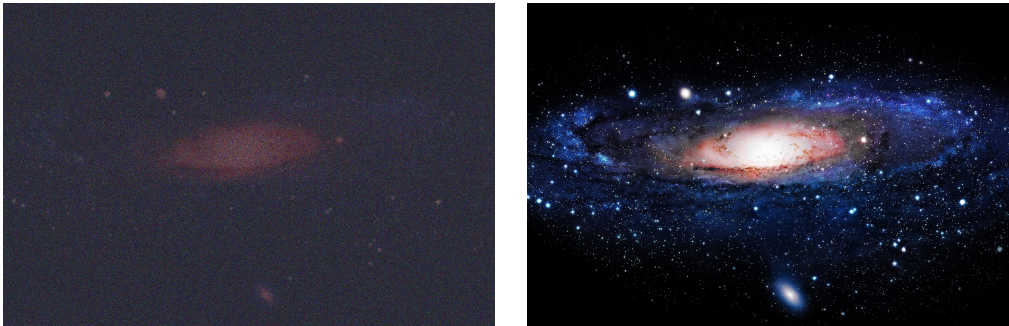

Figure 4: Recovered Galaxy images after 100 gradient iterations of TAF (Left); and of RAF (Right).

## 5    Conclusion

This paper developed a linear-time algorithm called RAF for solving systems of random quadratic equations. Our procedure consists of two stages: a weighted maximal correlation initializer attainable with a few power or Lanczos iterations, and a sequence of scalable reweighted gradient refinements for a nonconvex nonsmooth LS loss function. It was demonstrated that RAF achieves the optimal sample and computational complexity. Judicious numerical tests showcase its superior performance over state-of-the-art alternatives. Empirically, RAF solves a set of random quadratic equations with high probability so long as a unique solution exists. Promising extensions include studying robust and/or sparse phase retrieval and matrix recovery via (stochastic) reweighted amplitude flow counterparts, and in particular exploiting the power of (re)weighting regularization techniques to enable more general nonconvex optimization such as training deep neural networks [18].

### Acknowledgments

G. Wang and G. B. Giannakis were partially supported by NSF grants 1500713 and 1514056. Y. Saad was partially supported by NSF grant 1505970. J. Chen was partially supported by the National Natural Science Foundation of China grants U1509215, 61621063, and the Program for Changjiang Scholars and Innovative Research Team in University (IRT1208).

## Footnotes

[1]It is out of the scope of the present paper to explain the meaning of generic vectors, whereas interested readers are referred to [1].

[2] $f(m) = \mathcal{O}(g(m))$ means that there exists a constant $C > 0$ such that $|f(m)| \leq C|g(m)|$.

[3] Downloaded from `http://pics-about-space.com/milky-way-galaxy`.

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
