[Supplementary Material]

# Supplementary Material for "Solving Most Systems of Random Quadratic Equations"

G. Wang[*,★], G. B. Giannakis[*], Y. Saad[†], and J. Chen[★]

[*]Digital Tech. Center & ECE Dept., U. of Minnesota, Mlps., MN 55455, USA

[★]Key Lab of Intelligent Control and Decision of Complex Systems, Beijing Institute of Technology, Beijing 100081, China

[†] CSE Dept., U. of Minnesota, Mlps., MN 55455, USA

{gangwang, georgios, saad}@umn.edu; chenjie@bit.edu.cn.

## 1 Algorithm and main theorem

For analytical concreteness, our technical results in the following are based on the real Gaussian model. For readability and future reference, we begin with repeating our model, notation, and main assumptions: Consider first the noiseless case

$$\psi_i = |\boldsymbol{a}_i^* \boldsymbol{x}|, \quad 1 \leq i \leq m \tag{1}$$

in which $\boldsymbol{x} \in \mathbb{R}^n$ is the wanted unknown signal, and the sampling/feature vectors $\{\boldsymbol{a}_i \in \mathbb{R}^n\}_{1 \leq i \leq m}$ are drawn independently and identically from the $n$-dimensional real Gaussian distribution, i.e., i.i.d. $\boldsymbol{a}_i \sim \mathcal{N}(\boldsymbol{0}, \boldsymbol{I}_n)$. This is also referred to as the real Gaussian model in the literature.

For notational convenience, let $\boldsymbol{A} := [\boldsymbol{a}_1 \cdots \boldsymbol{a}_m]^*$, and $\boldsymbol{\psi} := [\psi_1 \cdots \psi_m]^*$. Assume that the (phaseless) quadratic system (1) admits a unique solution, which indeed holds with high probability as long as $m \geq 2n-1$ generic measurements are taken [1]. Throughout our subsequent analysis, fix $\boldsymbol{x}$ to be any solution of the given system in (1). If $\boldsymbol{x}$ obeys the system in (1), so does $-\boldsymbol{x}$; i.e., the solution set in the real case is $\{\boldsymbol{x}, -\boldsymbol{x}\}$. We focus on $\boldsymbol{x}$ without loss of generality, rather than $-\boldsymbol{x}$. Introduce the notion of Euclidean distance of any estimate $\boldsymbol{z}$ to the solution set: $\mathrm{dist}(\boldsymbol{z}, \boldsymbol{x}) := \min\{\|\boldsymbol{z}+\boldsymbol{x}\|, \|\boldsymbol{z}-\boldsymbol{x}\|\}$ for real signals; and $\mathrm{dist}(\boldsymbol{z}, \boldsymbol{x}) := \min_{\phi \in [0, 2\pi)} \|\boldsymbol{z} - \boldsymbol{x}e^{i\phi}\|$ for complex ones [2]. Define the indistinguishable global phase factor for real-valued signals [2]

$$\phi(\boldsymbol{z}) := \begin{cases} 0, & \|\boldsymbol{z} - \boldsymbol{x}\| \leq \|\boldsymbol{z} + \boldsymbol{x}\| \\ \pi, & \text{otherwise.} \end{cases} \tag{2}$$

Hereafter, we always assume $\phi(\boldsymbol{z}) = 0$; and $\boldsymbol{z}$ is replaced by $e^{-j\phi(\boldsymbol{z})}\boldsymbol{z}$ otherwise. For simplicity of presentation however, the constant phase adaptation term will be dropped whenever it is clear from the context.

Algorithm 1 and Theorem 1 in the paper are repeated next for future reference.

**Theorem 1** (Exact recovery). *Consider $m$ noiseless measurements $\boldsymbol{\psi} = |\boldsymbol{A}\boldsymbol{x}|$ for an arbitrary $\boldsymbol{x} \in \mathbb{R}^n$. If the data size $m \geq c_0|\mathcal{S}| \geq c_1 n$ and the step size $\mu \leq \mu_0$ , then with probability at least $1 - c_3 e^{-c_2 m}$, the reweighted amplitude flow's estimates $\boldsymbol{z}^t$ in Algorithm 1 obey*

$$\mathrm{dist}(\boldsymbol{z}^t, \boldsymbol{x}) \leq \frac{1}{10}(1 - \nu)^t \|\boldsymbol{x}\|, \quad t = 0, 1, \ldots \tag{5}$$

---

**Algorithm 1** Reweighted Amplitude Flow

---

1: **Input:** Data $\{(\boldsymbol{a}_i; \psi_i)\}_{1 \le i \le m}$; maximum no. of iterations $T$; step size $\mu^t = 2/6$ for real/complex Gaussian model; weighting parameters $|\mathcal{S}| = \lfloor 3m/13 \rfloor$, $\beta_i = 10$, and $\gamma = 0.5$.

2: **Obtain** $\mathcal{S}$ to include indices associated with the $|\mathcal{S}|$ largest entries among $\{\psi_i\}_{1 \le i \le m}$.

3: **Initialize** $\boldsymbol{z}_0 := \sqrt{\sum_{i=1}^{m} \psi_i^2/m} \, \tilde{\boldsymbol{z}}_0$ with $\tilde{\boldsymbol{z}}_0$ being the unit principal eigenvector of

$$\boldsymbol{Y} := \frac{1}{m} \sum_{i=1}^{m} w_i^0 \boldsymbol{a}_i \boldsymbol{a}_i^*, \quad \text{with} \quad w_i^0 := \begin{cases} \psi_i^\gamma, & i \in \mathcal{S} \subseteq \mathcal{M} \\ 0, & \text{otherwise} \end{cases}. \tag{3}$$

4: **Loop: for** $t = 0$ **to** $T-1$

$$\boldsymbol{z}^{t+1} = \boldsymbol{z}^t - \frac{\mu^t}{m} \sum_{i=1}^{m} w_i^t \left( \boldsymbol{a}_i^* \boldsymbol{z}^t - \psi_i \frac{\boldsymbol{a}_i^* \boldsymbol{z}^t}{|\boldsymbol{a}_i^* \boldsymbol{z}^t|} \right) \boldsymbol{a}_i \tag{4}$$

where $w_i^t := \frac{|\boldsymbol{a}_i^* \boldsymbol{z}^t|/\psi_i}{|\boldsymbol{a}_i^* \boldsymbol{z}^t|/\psi_i + \beta_i}$ for all $1 \le i \le m$.

5: **Output:** $\boldsymbol{z}^T$.

---

where $c_0$, $c_1$, $c_2$, $c_3 > 0$, $0 < \nu < 1$, and $\mu_0 > 0$ are numerical constants depending on the choice of algorithmic parameters $|\mathcal{S}|$, $\beta$, $\gamma$, and $\mu$.

# 2  Proofs

To prove Theorem 1, this section establishes a few lemmas and the main ideas, while technical details are posponed to the Appendix. Relative to the state-of-the-art [2], [5], [11], [10], [8], both the weighted maximal correlation initialization and the reweighted objective function in this paper are novel, and hence, new proof techniques to handle the reweighting, nonsmoothness, and nonconvexity are required. Nonetheless, part of the proof is built upon those in [2], [11], [10], [4].

The proof of Theorem 1 comprises two independent parts: Section 2.1 demonstrates the theoretical performance of the proposed initialization, which essentially achieves any given constant relative error as soon as the number of equations is on the order of the number of unknowns, that is, $m \ge c_1 n$ for some constant $c_1 > 0$. Under such a sample complexity, Section 2.2 shows that RAF converges to the true signal $\boldsymbol{x}$ exponentially fast whenever the initial estimate enjoys a small constant relative error, namely, $\text{dist}(\boldsymbol{z}^0, \boldsymbol{x}) \le \rho\|\boldsymbol{x}\|$ as in (5).

## 2.1  Weighted maximal correlation initialization

This section is devoted to establishing theoretical guarantees for the proposed weighted maximal correlation initialization, which is summarized in the following proposition. An alternative approach may be found in [6].

**Proposition 1.** *For arbitrary $\boldsymbol{x} \in \mathbb{R}^n$, consider the noiseless measurements $\psi_i = |\boldsymbol{a}_i^* \boldsymbol{x}|$, $1 \le i \le m$. If $m \ge c_0|\mathcal{S}| \ge c_1 n$, then with probability exceeding $1 - c_3' \mathrm{e}^{-c_2' m}$, the initialization $\boldsymbol{z}^0$ returned by the weighted maximal correlation method in Step 3 of Algorithm 1 satisfies*

$$\text{dist}(\boldsymbol{z}^0, \boldsymbol{x}) \le \rho\|\boldsymbol{x}\| \tag{6}$$

for $\rho = 1/10$ or any sufficiently small positive number. Here, $c_0$, $c_1$, $c_2'$, $c_3' > 0$ are certain universal constants.

Due to the homogeneity of (6) in $\boldsymbol{x}$, it suffices to prove the results for the case of $\|\boldsymbol{x}\| = 1$. Assume first that the norm $\|\boldsymbol{x}\| = 1$ is perfectly known, and $\boldsymbol{z}^0$ has already been scaled such that $\|\boldsymbol{z}^0\| = 1$. At the end of this section, this approximation error between the actually employed norm estimate $\sqrt{\sum_{i=1}^m y_i/m}$ found based on the strong law of large numbers and the unknown norm $\|\boldsymbol{x}\| = 1$ will be taken care of. For i.i.d. Gaussian random design vectors $\boldsymbol{a}_i \sim \mathcal{N}(\boldsymbol{0}, \boldsymbol{I}_n)$ and any unit vector $\boldsymbol{x}$, there exists an orthogonal transformation denoted by $\boldsymbol{U} \in \mathbb{R}^{n \times n}$ such that $\boldsymbol{x} = \boldsymbol{U}\boldsymbol{e}_1$. Since

$$|\langle \boldsymbol{a}_i, \boldsymbol{x} \rangle|^2 = |\langle \boldsymbol{a}_i, \boldsymbol{U}\boldsymbol{e}_1 \rangle|^2 = |\langle \boldsymbol{U}^* \boldsymbol{a}_i, \boldsymbol{e}_1 \rangle|^2 \overset{d}{=} |\langle \boldsymbol{a}_i, \boldsymbol{e}_1 \rangle|^2 \tag{7}$$

where $\overset{d}{=}$ means random quantities on both sides of the equality enjoy the same distribution, it is thus without loss of generality to assume $\boldsymbol{x} = \boldsymbol{e}_1$.

Since the norm $\|\boldsymbol{x}\| = 1$ is assumed known, the weighted maximal correlation initialization in Step 3 of Algorithm 1 finds the initial estimate $\boldsymbol{z}^0 = \tilde{\boldsymbol{z}}^0$ (the scaling factor is the exactly known norm 1 in this case) as the principal eigenvector of

$$\boldsymbol{Y} := \frac{1}{|\mathcal{S}|} \boldsymbol{B}^* \boldsymbol{B} = \frac{1}{|\mathcal{S}|} \sum_{i \in \mathcal{S}} \psi_i^\gamma \boldsymbol{a}_i \boldsymbol{a}_i^* \tag{8}$$

where $\boldsymbol{B} := \left[ \psi_i^{\gamma/2} \boldsymbol{a}_i \right]_{i \in \mathcal{S}}$ is an $|\mathcal{S}| \times n$ matrix, and $\mathcal{S} \subsetneq \{1, 2, \ldots, m\}$ includes the indices of the $|\mathcal{S}|$ largest entities among all modulus data $\{\psi_i\}_{1 \le i \le m}$. The following result is key to proving Proposition 1, whose proof is postponed to Section A.1 for readability.

**Lemma 1.** *Consider $m$ noiseless measurements $\psi_i = |\boldsymbol{a}_i^* \boldsymbol{x}|$, $1 \le i \le m$. For arbitrary $\boldsymbol{x} \in \mathbb{R}^n$ of unity norm, the next result holds for all unit vectors $\boldsymbol{u} \in \mathbb{R}^n$ perpendicular to the vector $\boldsymbol{x}$, namely, for all vectors $\boldsymbol{u} \in \mathbb{R}^n$ obeying $\boldsymbol{u}^* \boldsymbol{x} = 0$ and $\|\boldsymbol{u}\| = 1$:*

$$\frac{1}{2} \left\| \boldsymbol{x}\boldsymbol{x}^* - \boldsymbol{z}^0(\boldsymbol{z}^0)^* \right\|_F^2 \le \frac{\|\boldsymbol{B}\boldsymbol{u}\|^2}{\|\boldsymbol{B}\boldsymbol{x}\|^2} \tag{9}$$

*where $\boldsymbol{z}^0 = \tilde{\boldsymbol{z}}^0$ is given by*

$$\tilde{\boldsymbol{z}}^0 := \arg \max_{\|\boldsymbol{z}\|=1} \frac{1}{|\mathcal{S}|} \boldsymbol{z}^* \boldsymbol{B}^* \boldsymbol{B}\boldsymbol{z}. \tag{10}$$

In the sequel, we start proving Proposition 1. The first step consists in upper-bounding the term on the right-hand-side of (9). To be specific, the task involves upper bounding its numerator term, and lower bounding its denominator term, which are summarized in Lemma 2 and Lemma 3, whose proofs are deferred to Section A.2 and Section A.3, accordingly.

**Lemma 2.** *In the setting of Lemma 1, if $|\mathcal{S}|/n \ge c_4$, then the next*

$$\|\boldsymbol{B}\boldsymbol{u}\|^2 \le 1.01 \sqrt{2^\gamma/\pi} \, \Gamma(\gamma+1/2) |\mathcal{S}| \tag{11}$$

*holds with probability at least $1 - 2\mathrm{e}^{-c_K n}$, where $c_4$ and $c_K$ are some absolute constants.*

**Lemma 3.** *In the setting of Lemma 1, the following holds with probability exceeding $1 - \mathrm{e}^{-c_2' m}$:*

$$\|\boldsymbol{B}\boldsymbol{x}\|^2 \ge 0.99 |\mathcal{S}| \left[ 1 + \log(m/|\mathcal{S}|) \right] \ge 0.99 \cdot 1.14^\gamma |\mathcal{S}| \left[ 1 + \log(m/|\mathcal{S}|) \right] \tag{12}$$

*provided that $m \ge c_0 |\mathcal{S}| \ge c_1 n$ for some absolute constants $c_0$, $c_1$, $c_2' > 0$.*

Putting the upper bound (11) and the lower bound (12) together, one arrives at

$$\frac{\left\|\boldsymbol{B}\boldsymbol{u}\right\|^2}{\left\|\boldsymbol{B}\boldsymbol{x}\right\|^2} \leq \frac{C}{1 + \log(m/|\mathcal{S}|)} \triangleq \kappa \tag{13}$$

where the constant $C := 1.02 \cdot 1.14^{-\gamma}\sqrt{2^\gamma/\pi}\,\Gamma(\gamma+1/2)$ and which holds with probability at least $1-2\mathrm{e}^{-c_K n}-\mathrm{e}^{-c_2' m}$, with the proviso that $m \geq c_0|\mathcal{S}| \geq c_1 n$ for some universal constants $c_0, c_1, c_2', c_K > 0$. Since $m \geq c_1 n$, it is without loss of generality to rewrite the probability as $1-c_3'\mathrm{e}^{-c_2' m}$ for numerical constants $c_2', c_3' > 0$. To have a sense of the size of the constant $C$, taking our default value $\gamma = 0.5$ for instance gives rise to $C = 0.7854$.

Observe that the bound $\kappa$ in (13) can be made arbitrarily small through taking sufficiently large $m/|\mathcal{S}|$ values (while maintaining $|\mathcal{S}|/n$ large enough as well based on Lemma 3). With no loss of generality, let us work with $\kappa := 0.001$ in the following.

The wanted upper bound on the distance between the initialization $\boldsymbol{z}^0$ and the truth $\boldsymbol{x}$ can be obtained based upon similar arguments found in [2, Section 7.8], which are detailed as follows. For unit vectors $\boldsymbol{x}$ and $\boldsymbol{z}^0$, recall from (38) that

$$|\boldsymbol{x}^*\boldsymbol{z}^0|^2 = \cos^2\theta = 1 - \sin^2\theta \geq 1 - \kappa, \tag{14}$$

where $0 \leq \theta \leq \pi/2$ denotes the angle between the spaces spanned by $\boldsymbol{z}^0$ and $\boldsymbol{x}$, therefore

$$\begin{aligned}
\mathrm{dist}^2(\boldsymbol{z}^0, \boldsymbol{x}) &\leq \|\boldsymbol{z}^0\|^2 + \|\boldsymbol{x}\|^2 - 2|\boldsymbol{x}^*\boldsymbol{z}^0| \\
&\leq \left(2 - 2\sqrt{1-\kappa}\right)\|\boldsymbol{x}\|^2 \\
&\approx \kappa\,\|\boldsymbol{x}\|^2.
\end{aligned} \tag{15}$$

As discussed prior to Lemma 1, the exact norm $\|\boldsymbol{x}\| = 1$ is typically not known, and one often scales the unit directional vector found in (10) by the norm estimate $\sqrt{\sum_{i=1}^m \psi_i^2/m}$. Next the approximation error between the estimated norm $\|\boldsymbol{z}^0\| = \sqrt{\sum_{i=1}^m \psi_i^2/m}$ and the true norm $\|\boldsymbol{x}\| = 1$ is accounted for. Recall from (10) that the direction of $\boldsymbol{x}$ is estimated to be $\tilde{\boldsymbol{z}}^0$ (of unity norm). Using similar results in [2, Lemma 7.8 and Section 7.8], the following holds with high probability as long as the ratio $m/n$ exceeds some numerical constant

$$\|\boldsymbol{z}^0 - \tilde{\boldsymbol{z}}^0\| = |\|\boldsymbol{z}^0\| - 1| \leq (1/20)\|\boldsymbol{x}\|. \tag{16}$$

Taking the inequalities in (15) and (16) together, it is safe to conclude that

$$\mathrm{dist}(\boldsymbol{z}^0, \boldsymbol{x}) \leq \|\boldsymbol{z}^0 - \tilde{\boldsymbol{z}}^0\| + \mathrm{dist}(\tilde{\boldsymbol{z}}^0, \boldsymbol{x}) \leq (1/10)\|\boldsymbol{x}\| \tag{17}$$

which validates that the initialization satisfies the relative error $\mathrm{dist}(\boldsymbol{z}^0, \boldsymbol{x})/\|\boldsymbol{x}\| \leq 1/10$ for any $\boldsymbol{x} \in \mathbb{R}^n$ with probability at least $1 - c_3'\mathrm{e}^{-c_2' m}$, provided that $m \geq c_0|\mathcal{S}| \geq c_1 n$ holds for some numerical constants $c_0, c_1, c_2', c_3' > 0$.

## 2.2 Exact Phase Retrieval from Noiseless Data

It has been demonstrated that the initial estimate $\boldsymbol{z}^0$ obtained by means of the weighted maximal correlation initialization strategy, i.e., Step 3 of Algorithm 1, has at most a constant relative error

to the globally optimal solution $\boldsymbol{x}$, i.e., $\mathrm{dist}(\boldsymbol{z}^0, \boldsymbol{x}) \leq (1/10)\|\boldsymbol{x}\|$. In the sequel, we demonstrate that starting from such an initial point, our RAF iterates converge with high probability at an exponential rate to the global optimum $\boldsymbol{x}$, namely, $\mathrm{dist}(\boldsymbol{z}^t, \boldsymbol{x}) \leq (1/10)c^t\|\boldsymbol{x}\|$ for some constant $0 < c < 1$ depending on the step size $\mu > 0$, the weighting parameter $\beta_i \equiv \beta$, and the data $\{(\boldsymbol{a}_i; \psi_i)\}_{1 \leq i \leq m}$. This is indeed concerned with the second part of Theorem 1. To this end, it suffices to show that the iterative updates of RAF, namely, Step 4 of Algorithm 1 is locally contractive within a small enough neighboring region of the truth $\boldsymbol{x}$. Recall first that RAF's update is based on the reweighted gradient

$$
\begin{aligned}
\nabla \ell_{\mathrm{rw}}(\boldsymbol{z}) &:= \frac{1}{m} \sum_{i=1}^{m} w_i \left( \boldsymbol{a}_i^* \boldsymbol{z} - \psi_i \frac{\boldsymbol{a}_i^* \boldsymbol{z}}{|\boldsymbol{a}_i^* \boldsymbol{z}|} \right) \boldsymbol{a}_i \\
&= \frac{1}{m} \sum_{i=1}^{m} w_i \left( \boldsymbol{a}_i^* \boldsymbol{z} - |\boldsymbol{a}_i^* \boldsymbol{x}| \frac{\boldsymbol{a}_i^* \boldsymbol{z}}{|\boldsymbol{a}_i^* \boldsymbol{z}|} \right) \boldsymbol{a}_i
\end{aligned}
\tag{18}
$$

for judiciously designed weights

$$
w_i = \frac{1}{1 + \beta / (|\boldsymbol{a}_i^* \boldsymbol{z}| / |\boldsymbol{a}_i^* \boldsymbol{x}|)}, \quad 1 \leq i \leq m
\tag{19}
$$

in which the dependence of pertinent terms on the iterate index $t$ is ignored for notational brevity.

**Proposition 2** (Local error contraction). *For arbitrary $\boldsymbol{x} \in \mathbb{R}^n$, consider $m$ noise-free measurements $\psi_i = |\boldsymbol{a}_i^* \boldsymbol{x}|$, $1 \leq i \leq m$. There exist some numerical constants $c_1$, $c_2''$, $c_3'' > 0$, and $0 < \nu < 1$ such that the following holds with probability exceeding $1 - c_3'' \mathrm{e}^{-c_2'' m}$*

$$
\mathrm{dist}^2(\boldsymbol{z} - \mu \nabla \ell_{\mathrm{rw}}(\boldsymbol{z}), \boldsymbol{x}) \leq (1 - \nu)\mathrm{dist}^2(\boldsymbol{z}, \boldsymbol{x})
\tag{20}
$$

*for all $\boldsymbol{x}$, $\boldsymbol{z} \in \mathbb{R}^n$ obeying $\mathrm{dist}(\boldsymbol{z}, \boldsymbol{x}) \leq (1/10)\|\boldsymbol{x}\|$, provided that $m \geq c_1 n$ and that the constant step size $0 < \mu \leq \mu_0$, where $\mu_0$ is some numerical constant depending on the weighting parameter $\beta > 0$ and the data $\{(\boldsymbol{a}_i; \psi_i)\}_{1 \leq i \leq m}$.*

Proposition 2 essentially illustrates that the distance of RAF's successive iterates to the global optimum $\boldsymbol{x}$ decreases monotonically once the algorithm's iterate $\boldsymbol{z}^t$ enters a relatively small neighboring region around the truth $\boldsymbol{x}$. This small-size neighborhood is commonly known as the *basin of attraction*, and has been widely discussed in recent nonconvex optimization works; see, for instance, [2], [11], [10]. Expressed differently, RAF's iterates will stay within the region and will be attracted towards the global optimum $\boldsymbol{x}$ exponentially fast as soon as it lands within the basin of attraction. To substantiate Proposition 2, recall the useful analytical tool of the local regularity condition [2], which plays a key role in establishing geometric or linear convergence to the global optimum for nonconvex optimization schemes [2], [8], [11], [10], [9].

For RAF in the present work, the reweighted gradient $\nabla \ell_{\mathrm{rw}}(\boldsymbol{z})$ defined in (18) is said to obey the local regularity condition, or $\mathrm{LRC}(\mu, \lambda, \epsilon)$ for some constant $\lambda > 0$, if the following inequality

$$
\langle \nabla \ell_{\mathrm{rw}}(\boldsymbol{z}), \boldsymbol{h} \rangle \geq \frac{\mu}{2} \|\nabla \ell_{\mathrm{rw}}(\boldsymbol{z})\|^2 + \frac{\lambda}{2} \|\boldsymbol{h}\|^2
\tag{21}
$$

holds for all $\boldsymbol{z} \in \mathbb{R}^n$ such that $\|\boldsymbol{h}\| = \|\boldsymbol{z} - \boldsymbol{x}\| \leq \epsilon \|\boldsymbol{x}\|$ for some constant $0 < \epsilon < 1$, where the ball given by $\|\boldsymbol{z} - \boldsymbol{x}\| \leq \epsilon \|\boldsymbol{x}\|$ is the so-termed *basin of attraction*.

Realizing $\boldsymbol{h} := \boldsymbol{z} - \boldsymbol{x}$, some algebraic manipulations in conjunction with the regularity condition

in (21) gives rise to

$$
\begin{aligned}
\operatorname{dist}^2(\boldsymbol{z} - \mu\nabla\ell_{\mathrm{rw}}(\boldsymbol{z}),\, \boldsymbol{x}) &= \|\boldsymbol{z} - \mu\nabla\ell_{\mathrm{rw}}(\boldsymbol{z}) - \boldsymbol{x}\|^2 \\
&= \|\boldsymbol{h}\|^2 - 2\mu\,\langle \boldsymbol{h}, \nabla\ell_{\mathrm{rw}}(\boldsymbol{z})\rangle + \|\mu\nabla\ell_{\mathrm{rw}}(\boldsymbol{z})\|^2 \qquad (22)\\
&\le \|\boldsymbol{h}\|^2 - 2\mu\left(\frac{\mu}{2}\|\nabla\ell_{\mathrm{rw}}(\boldsymbol{z})\|^2 + \frac{\lambda}{2}\|\boldsymbol{h}\|^2\right) + \|\mu\nabla\ell_{\mathrm{rw}}(\boldsymbol{z})\|^2 \\
&= (1 - \lambda\mu)\|\boldsymbol{h}\|^2 = (1 - \lambda\mu)\operatorname{dist}^2(\boldsymbol{z},\, \boldsymbol{x}) \qquad (23)
\end{aligned}
$$

for all points $\boldsymbol{z}$ adhering to the relative error $\|\boldsymbol{h}\| \le \epsilon\|\boldsymbol{x}\|$. It is self-evident that if the regularity condition $\mathrm{LRC}(\mu, \lambda, \epsilon)$ can be established for RAF, our ultimate goal of proving the local error contraction in (20) follows straightforwardly upon letting $\nu := \lambda\mu$.

### 2.2.1 Proof of the local regularity condition in (21)

The first step of proving the local regularity condition in (21) is to control the size of the reweighted gradient $\nabla\ell_{\mathrm{rw}}(\boldsymbol{z})$, i.e., to upper bound the last term in (22). Note that the reweighted gradient can be rewritten in a compact matrix-vector representation

$$
\nabla\ell_{\mathrm{rw}}(\boldsymbol{z}) = \frac{1}{m}\sum_{i=1}^{m} w_i\left(\boldsymbol{a}_i^*\boldsymbol{z} - |\boldsymbol{a}_i^*\boldsymbol{x}|\frac{\boldsymbol{a}_i^*\boldsymbol{z}}{|\boldsymbol{a}_i^*\boldsymbol{z}|}\right)\boldsymbol{a}_i \triangleq \frac{1}{m}\operatorname{diag}(\boldsymbol{w})\boldsymbol{A}\boldsymbol{v} \qquad (24)
$$

where $\operatorname{diag}(\boldsymbol{w}) \in \mathbb{R}^{n\times n}$ is a diagonal matrix holding entries of $\boldsymbol{w} := [w_1 \ \cdots \ w_m]^* \in \mathbb{R}^m$ on its main diagonal, and $\boldsymbol{v} := [v_1 \ \cdots \ v_m]^* \in \mathbb{R}^m$ with $v_i := \boldsymbol{a}_i^*\boldsymbol{z} - |\boldsymbol{a}_i^*\boldsymbol{x}|\frac{\boldsymbol{a}_i^*\boldsymbol{z}}{|\boldsymbol{a}_i^*\boldsymbol{z}|}$. Using the definition of induced matrix 2-norm (or the matrix spectral norm), it is easy to check that

$$
\begin{aligned}
\|\nabla\ell_{\mathrm{rw}}(\boldsymbol{z})\| &= \left\|\frac{1}{m}\operatorname{diag}(\boldsymbol{w})\boldsymbol{A}\boldsymbol{v}\right\| \\
&\le \frac{1}{m}\|\operatorname{diag}(\boldsymbol{w})\| \cdot \|\boldsymbol{A}\| \cdot \|\boldsymbol{v}\| \\
&\le \frac{1 + \delta'}{\sqrt{m}}\|\boldsymbol{v}\| \qquad (25)
\end{aligned}
$$

where we have used the inequalities $\|\operatorname{diag}(\boldsymbol{w})\| \le 1$ due to $w_i \le 1$ for all $1 \le i \le m$, and $\|\boldsymbol{A}\| \le (1 + \delta')\sqrt{m}$ for some constant $\delta' > 0$ according to [7, Theorem 5.32], provided that $m/n$ is sufficiently large.

The task therefore remains to bound $\|\boldsymbol{v}\|$ in (25), which is addressed next. To this end, notice that

$$
\begin{aligned}
\|\boldsymbol{v}\|^2 = \sum_{i=1}^{m}\left(\boldsymbol{a}_i^*\boldsymbol{z} - |\boldsymbol{a}_i^*\boldsymbol{x}|\frac{\boldsymbol{a}_i^*\boldsymbol{z}}{|\boldsymbol{a}_i^*\boldsymbol{z}|}\right)^2 &\le \sum_{i=1}^{m}\left(|\boldsymbol{a}_i^*\boldsymbol{z}| - |\boldsymbol{a}_i^*\boldsymbol{x}|\right)^2 \\
&\le \sum_{i=1}^{m}\left(\boldsymbol{a}_i^*\boldsymbol{z} - \boldsymbol{a}_i^*\boldsymbol{x}\right)^2 \\
&= \sum_{i=1}^{m}(\boldsymbol{a}_i^*\boldsymbol{h})^2 \le (1 + \delta'')^2 m\|\boldsymbol{h}\|^2 \qquad (26)
\end{aligned}
$$

for some numerical constant $\delta'' > 0$, where the last can be obtained using [3, Lemma 3.1] and which holds with probability at least $1 - \mathrm{e}^{-c_2 m}$ as long as $m > c_1 n$ holds true.

Combing the results in (25) and (26) and taking $\delta > 0$ larger than the constant $(1 + \delta')(1 + \delta'') - 1$, the size of the reweighted gradient $\nabla \ell_{\mathrm{rw}}(\boldsymbol{z})$ can be bounded as follows

$$\|\nabla \ell_{\mathrm{rw}}(\boldsymbol{z})\| \leq (1 + \delta)\|\boldsymbol{h}\| \tag{27}$$

which holds with probability $1 - \mathrm{e}^{-c_2 m}$, with a proviso that $m/n$ exceeds some numerical constant $c_1 > 0$. This result indeed suggests that the reweighted gradient of the objective function $L(\boldsymbol{z})$ or the search direction employed in RAF algorithm is well behaved, implying that the function value along the iterates does not change too much.

In order to prove the LRC, it suffices to show that the reweighted gradient $\nabla \ell_{\mathrm{rw}}(\boldsymbol{z})$ ensures sufficient descent, that is, there exists a numerical constant $c > 0$ such that along the search direction $\nabla \ell_{\mathrm{rw}}(\boldsymbol{z})$ the following uniform lower bound holds

$$\langle \nabla \ell_{\mathrm{rw}}(\boldsymbol{z}), \boldsymbol{h} \rangle \geq c\|\boldsymbol{h}\|^2 \tag{28}$$

which will be addressed in this section. Formally, this can be summarized in the following proposition, whose proof is provided in Appendix A.4.

**Proposition 3.** *Fixing any sufficiently small constant $\epsilon > 0$, consider the noise-free measurements $\psi_i = |\boldsymbol{a}_i^* \boldsymbol{x}|$, $1 \leq i \leq m$. There exist some numerical constants $c_1$, $c_2'$, $c_3' > 0$ such that the following holds with probability at least $1 - c_3' \mathrm{e}^{-c_2' m}$:*

$$\langle \boldsymbol{h}, \nabla \ell_{\mathrm{rw}}(\boldsymbol{z}) \rangle \geq \left[ \frac{1 - \zeta_1 - \epsilon}{1 + \beta(1 + \eta)} - 2(\zeta_2 + \epsilon) - \frac{2(0.1271 - \zeta_2 + \epsilon)}{1 + \beta/k} \right] \|\boldsymbol{h}\|^2 \tag{29}$$

*for all $\boldsymbol{x}$, $\boldsymbol{z} \in \mathbb{R}^n$ obeying $\|\boldsymbol{h}\| \leq \frac{1}{10}\|\boldsymbol{x}\|$, provided that $m/n$ is large enough, and that $\beta > 0$ is small enough.*

Taking the results in (29) and (27) together back to (21), one concludes that the local regularity condition holds for $\mu$ and $\lambda$ obeying the following

$$\frac{1 - \zeta_1 - \epsilon}{1 + \beta(1 + \eta)} - 2(\zeta_2 + \epsilon) - \frac{2(0.1271 - \zeta_2 + \epsilon)}{1 + \beta/k} \geq \frac{\mu}{2}(1 + \delta)^2 + \frac{\lambda}{2}. \tag{30}$$

For instance, take $\beta = 2$, $k = 5$, $\eta = 0.5$, and $\epsilon = 0.001$, we have $\zeta_1 = 0.8897$ and $\zeta_2 = 0.0213$, thus asserting that $\langle \ell_{\mathrm{rw}}(\boldsymbol{z}), \boldsymbol{h} \rangle \geq 0.1065\|\boldsymbol{h}\|^2$. Setting further $\delta = 0.001$ leads to

$$0.1065 \geq 0.501\mu + 0.5\lambda \tag{31}$$

which concludes the proof of the local regularity condition in (21). Moreover, the local error contraction in (20) follows from substituting the local regularity condition into (23), hence validating Proposition 2.

# A  Proof details

By homogeneity, it suffices to work with the case where $\|\boldsymbol{x}\| = 1$.

## A.1  Proof of Lemma 1

It is easy to check that

$$\frac{1}{2}\left\|\boldsymbol{x}\boldsymbol{x}^* - \boldsymbol{z}^0(\boldsymbol{z}^0)^*\right\|_F^2 = \frac{1}{2}\|\boldsymbol{x}\|^4 + \frac{1}{2}\|\boldsymbol{z}^0\|^4 - |\boldsymbol{x}^*\boldsymbol{z}^0|^2$$
$$= 1 - |\boldsymbol{x}^*\boldsymbol{z}^0|^2$$
$$= 1 - \cos^2\theta \tag{32}$$

where $0 \le \theta \le \pi/2$ denotes the angle between the hyperplanes spanned by $\boldsymbol{x}$ and $\boldsymbol{z}^0$. Letting $(\boldsymbol{z}^0)^\perp \in \mathbb{R}^n$ be a unit vector orthogonal to $\boldsymbol{z}^0$ and have a nonnegative inner-product with $\boldsymbol{x}$, then $\boldsymbol{x}$ can be uniquely expressed as a linear communication of $\boldsymbol{z}^0$ and $(\boldsymbol{z}^0)^\perp$, yielding

$$\boldsymbol{x} = \boldsymbol{z}^0\cos\theta + (\boldsymbol{z}^0)^\perp\sin\theta. \tag{33}$$

Likewise, introduce the unit vector $\boldsymbol{x}^\perp$ to be orthogonal to $\boldsymbol{x}$ and to have a nonnegative inner-product with $(\boldsymbol{z}^0)^\perp$. Therefore, $\boldsymbol{x}^\perp$ can be uniquely written as

$$\boldsymbol{x}^\perp := -\boldsymbol{z}^0\sin\theta + (\boldsymbol{z}^0)^\perp\cos\theta. \tag{34}$$

Recall from (10) (after ignoring the normalization factor $1/|\mathcal{S}|$) that $\boldsymbol{z}^0$ is the solution to the principal component analysis (PCA) problem

$$\boldsymbol{z}^0 := \arg\max_{\|\boldsymbol{z}\|=1}\ \boldsymbol{z}^*\boldsymbol{B}^*\boldsymbol{B}\boldsymbol{z}. \tag{35}$$

Therefore, it holds that $\boldsymbol{B}^*\boldsymbol{B}\boldsymbol{z}^0 = \lambda_1\boldsymbol{z}^0$, where $\lambda_1 > 0$ is the largest eigenvalue of $\boldsymbol{B}^*\boldsymbol{B}$. Multiplying (33) and (34) by $\boldsymbol{B}$ from the left gives rise to

$$\boldsymbol{B}\boldsymbol{x} = \boldsymbol{B}\boldsymbol{z}^0\cos\theta + \boldsymbol{B}(\boldsymbol{z}^0)^\perp\sin\theta, \tag{36a}$$
$$\boldsymbol{B}\boldsymbol{x}^\perp = -\boldsymbol{B}\boldsymbol{z}^0\sin\theta + \boldsymbol{B}(\boldsymbol{z}^0)^\perp\cos\theta. \tag{36b}$$

Taking the 2-norm square of both sides in (36a) and (36b) yields

$$\|\boldsymbol{B}\boldsymbol{x}\|^2 = \|\boldsymbol{B}\boldsymbol{z}^0\|^2\cos^2\theta + \|\boldsymbol{B}(\boldsymbol{z}^0)^\perp\|^2\sin^2\theta, \tag{37a}$$
$$\|\boldsymbol{B}\boldsymbol{x}^\perp\|^2 = \|\boldsymbol{B}\boldsymbol{z}^0\|^2\sin^2\theta + \|\boldsymbol{B}(\boldsymbol{z}^0)^\perp\|^2\cos^2\theta, \tag{37b}$$

where the cross-terms disappear due to $(\boldsymbol{z}^0)^*\boldsymbol{B}^*\boldsymbol{B}(\boldsymbol{z}^0)^\perp = \lambda_1(\boldsymbol{z}^0)^*(\boldsymbol{z}^0)^\perp = 0$ according to the definition of $(\boldsymbol{z}^0)^\perp$.

With the relationships established in (37), construct now the following

$$\|\boldsymbol{B}\boldsymbol{x}\|^2\sin^2\theta - \|\boldsymbol{B}\boldsymbol{x}^\perp\|^2$$
$$= (\|\boldsymbol{B}\boldsymbol{z}^0\|^2\cos^2\theta + \|\boldsymbol{B}(\boldsymbol{z}^0)^\perp\|^2\sin^2\theta)\sin^2\theta - (\|\boldsymbol{B}\boldsymbol{z}^0\|^2\sin^2\theta + \|\boldsymbol{B}(\boldsymbol{z}^0)^\perp\|^2\cos^2\theta)$$
$$= (\|\boldsymbol{B}\boldsymbol{z}^0\|^2\cos^2\theta - \|\boldsymbol{B}\boldsymbol{z}^0\|^2 + \|\boldsymbol{B}(\boldsymbol{z}^0)^\perp\|^2\sin^2\theta)\sin^2\theta - \|\boldsymbol{B}(\boldsymbol{z}^0)^\perp\|^2\cos^2\theta$$
$$= (\|\boldsymbol{B}(\boldsymbol{z}^0)^\perp\|^2 - \|\boldsymbol{B}\boldsymbol{z}^0\|^2)\sin^4\theta - \|\boldsymbol{B}(\boldsymbol{z}^0)^\perp\|^2\cos^2\theta$$
$$\le 0$$

where $\boldsymbol{B}^*\boldsymbol{B} \succeq \boldsymbol{0}$, so $\|\boldsymbol{B}(\boldsymbol{z}^0)^\perp\|^2 - \|\boldsymbol{B}\boldsymbol{z}^0\|^2 \le 0$ holds for any unit vector $(\boldsymbol{z}^0)^\perp \in \mathbb{R}^n$ because $\boldsymbol{z}^0$ maximizes the term in (10), hence yielding

$$\sin^2\theta = 1 - \cos^2\theta \le \frac{\|\boldsymbol{B}\boldsymbol{x}^\perp\|^2}{\|\boldsymbol{B}\boldsymbol{x}\|^2}. \tag{38}$$

Plugging (32) into above, (9) can be proved by simply letting $\boldsymbol{u} = \boldsymbol{x}^\perp$.

## A.2 Proof of Lemma 2

Let $\{\boldsymbol{b}_i^*\}_{1 \leq i \leq |\mathcal{S}|}$ denote rows of $\boldsymbol{B} \in \mathbb{R}^{|\mathcal{S}| \times n}$, which are obtained by scaling rows of $\boldsymbol{A}_\mathcal{S} := \{\boldsymbol{a}_i^*\}_{i \in \mathcal{S}} \in \mathbb{R}^{|\mathcal{S}| \times n}$, the submatrix of $\boldsymbol{A}$ by the weights $\{\psi_i^\gamma\}_{i \in \mathcal{S}}$ accordingly [cf. (8)]. Since $\boldsymbol{x} = \boldsymbol{e}_1$, then $\boldsymbol{\psi} = |\boldsymbol{A}\boldsymbol{e}_1| = |\boldsymbol{A}_1|$, namely, the index set $\mathcal{S}$ depends only on the first column of $\boldsymbol{A}$, and is independent of the other columns of $\boldsymbol{A}$. In this direction, partition accordingly $\boldsymbol{A}^\mathcal{S} := [\boldsymbol{A}_1^\mathcal{S} \ \boldsymbol{A}_r^\mathcal{S}]$, where $\boldsymbol{A}_1^\mathcal{S} \in \mathbb{R}^{|\mathcal{S}| \times 1}$ denotes the first column of $\boldsymbol{A}^\mathcal{S}$, and $\boldsymbol{A}_r^\mathcal{S} \in \mathbb{R}^{|\mathcal{S}| \times (n-1)}$ collects the remaining ones. Likewise, partition $\boldsymbol{B} = [\boldsymbol{B}_1 \ \boldsymbol{B}_r]$ with $\boldsymbol{B}_1 \in \mathbb{R}^{|\mathcal{S}| \times 1}$ and $\boldsymbol{B}_r \in \mathbb{R}^{|\mathcal{S}| \times (n-1)}$. By the argument above, rows of $\boldsymbol{A}^\mathcal{S}$ are mutually independent, and they follow i.i.d. Gaussian distribution with mean $\boldsymbol{0}$ and covariance matrix $\boldsymbol{I}_{n-1}$. Furthermore, the weights $\psi_i^\gamma = |\boldsymbol{a}_i^* \boldsymbol{e}_1|^\gamma = |a_{i,1}|^\gamma$, $\forall i \in \mathcal{S}$ are also independent of the entries in $\boldsymbol{A}^\mathcal{S}$, where we recall that $\psi_{[m]} \leq \cdots \leq \psi_{[2]} \leq \psi_{[1]}$ represents the sorted sequence. As a consequence, rows of $\boldsymbol{B}_r$ are mutually independent of each other, and one can explicitly write its $i$-th row as $\boldsymbol{b}_{r,i} = |\boldsymbol{a}_{[i]}^* \boldsymbol{e}_1|^{\gamma/2} \boldsymbol{a}_{[i],\backslash 1} = |a_{[i],1}|^{\gamma/2} \boldsymbol{a}_{[i],\backslash 1}$, where $\boldsymbol{a}_{[i],\backslash 1} \in \mathbb{R}^{n-1}$ is obtained through removing the first entry of $\boldsymbol{a}_{[i]}$. It is easy to verify that $\mathbb{E}[\boldsymbol{b}_{r,i}] = \boldsymbol{0}$, and $\mathbb{E}[\boldsymbol{b}_{r,i} \boldsymbol{b}_{r,i}^*] = C_\gamma \boldsymbol{I}_{n-1}$, where the constant $C_\gamma := \sqrt{2^\gamma/\pi} \Gamma(\gamma+1/2) \|\boldsymbol{x}\|^\gamma = \sqrt{2^\gamma/\pi} \Gamma(\gamma+1/2)$, and $\Gamma(\cdot)$ is the Gamma function.

Given $\boldsymbol{x}^* \boldsymbol{x}^\perp = \boldsymbol{e}_1^* \boldsymbol{x}^\perp = 0$, one can write $\boldsymbol{x}^\perp = [0 \ \boldsymbol{r}^*]^*$ with any unit vector $\boldsymbol{r} \in \mathbb{R}^{n-1}$, hence

$$\|\boldsymbol{B}\boldsymbol{x}^\perp\|^2 = \|\boldsymbol{B}[0 \ \boldsymbol{r}^*]^*\|^2 = \|\boldsymbol{B}_r \boldsymbol{r}\|^2 \tag{39}$$

with independent subgaussian rows $\boldsymbol{b}_{r,i} = |a_{j,1}|^{\gamma/2} \boldsymbol{a}_{j,\backslash 1}$ if $0 \leq \gamma \leq 1$. Standard concentration results on the sum of random positive semi-definite matrices composed of independent non-isotropic subgaussian rows [7, Remark 5.40.1] assert that

$$\left\| \frac{1}{|\mathcal{S}|} \boldsymbol{B}_r^* \boldsymbol{B}_r - C_\gamma \boldsymbol{I}_{n-1} \right\| \leq \delta \tag{40}$$

holds with probability at least $1 - 2\mathrm{e}^{-c_K n}$ provided that $|\mathcal{S}|/n$ is larger than some positive constant. Here, $\delta > 0$ is a numerical constant that can take arbitrarily small values, and $c_K > 0$ is a constant depending on $\delta$. With no loss of generality, take $\delta := 0.01 C_\gamma$ in (40). For any unit vector $\boldsymbol{r} \in \mathbb{R}^{n-1}$, the following holds with probability at least $1 - 2\mathrm{e}^{-c_K n}$

$$\left\| \frac{1}{|\mathcal{S}|} \boldsymbol{r}^* \boldsymbol{B}_r^* \boldsymbol{B}_r \boldsymbol{r} - C_\gamma \boldsymbol{r}^* \boldsymbol{r} \right\| \leq \delta \boldsymbol{r}^* \boldsymbol{r} = \delta \tag{41}$$

or

$$\|\boldsymbol{B}_r \boldsymbol{r}\|^2 = \boldsymbol{r}^* \boldsymbol{B}_r^* \boldsymbol{B}_r \boldsymbol{r} \leq 1.01 C_\gamma |\mathcal{S}|. \tag{42}$$

Taking the last back to (39) confirms that

$$\|\boldsymbol{B}\boldsymbol{x}^\perp\|^2 \leq 1.01 C_\gamma |\mathcal{S}| \tag{43}$$

holds with probability at least $1 - 2\mathrm{e}^{-c_K n}$ if $|\mathcal{S}|/n$ exceeds some constant. Note that $c_K$ depends on the maximum subgaussian norm of the rows $\boldsymbol{b}_i$ in $\boldsymbol{B}_r$, and we assume without loss of generality $c_K \geq 1/2$. Therefore, one confirms that the numerator $\|\boldsymbol{B}\boldsymbol{u}\|^2$ in (9) is upper bounded via replacing $\boldsymbol{x}^\perp$ with $\boldsymbol{u}$ in (43).

## A.3 Proof of Lemma 3

This section is devoted to obtaining a meaningful lower bound for the denominator $\|\boldsymbol{B}\boldsymbol{x}\|^2$ in (12). Note first that

$$\|\boldsymbol{B}\boldsymbol{x}\|^2 = \sum_{i=1}^{|\mathcal{S}|} \|\boldsymbol{b}_i^* \boldsymbol{x}\|^2 = \sum_{i=1}^{|\mathcal{S}|} \psi_{[i]}^\gamma |\boldsymbol{a}_{[i]}^* \boldsymbol{x}|^2 = \sum_{i=1}^{|\mathcal{S}|} |\boldsymbol{a}_{[i]}^* \boldsymbol{x}|^{2+\gamma}.$$

Taking without loss of generality $\boldsymbol{x} = \boldsymbol{e}_1$, the term on the right side of the last equality reduces to

$$\|\boldsymbol{Bx}\|^2 = \sum_{i=1}^{|\mathcal{S}|} |a_{[i],1}|^{2+\gamma}. \tag{44}$$

Since $a_{[i],1}$ follows the standard normal distribution, the probability density function (pdf) of random variables $|a_{[i],1}|^{2+\gamma}$ can be given in closed form as

$$p(t) = \sqrt{\frac{2}{\pi}} \cdot \frac{1}{2+\gamma} t^{-\frac{1+\gamma}{2+\gamma}} e^{-\frac{1}{2}t^{\frac{2}{2+\gamma}}}, \quad t > 0 \tag{45}$$

which is rather complicated and whose cumulative density function (cdf) does not come in closed-form in general. Therefore, instead of dealing with the pdf in (45) directly, we shall take a different route by deriving a lower bound that is a bit looser yet suffices for our purpose, which is detailed as follows.

Since $|a_{[|\mathcal{S}|],1}| \leq \cdots \leq |a_{[2],1}| \leq |a_{[1],1}|$, then it holds for all $1 \leq i \leq |\mathcal{S}|$ that $|a_{[i],1}|^{2+\gamma} \geq |a_{[|\mathcal{S}|],1}|^{\gamma} a_{[i],1}^2$, therefore yielding

$$\|\boldsymbol{Bx}\|^2 = \sum_{i=1}^{|\mathcal{S}|} |a_{[i],1}|^{2+\gamma} \geq |a_{[|\mathcal{S}|],1}|^{\gamma} \sum_{i=1}^{|\mathcal{S}|} a_{[i],1}^2. \tag{46}$$

Hence, we next demonstrate that deriving a lower bound for $\|\boldsymbol{Bx}\|^2$ suffices to derive a lower bound for the summation on the right hand side above. The latter can be achieved by appealing to a result in [10, Lemma 3], which for completeness is included in the following.

**Lemma 4.** *For arbitrary unit vector $\boldsymbol{x} \in \mathbb{R}^n$, let $\psi_i = |\boldsymbol{a}_i^* \boldsymbol{x}|$, $1 \leq i \leq m$ be $m$ noiseless measurements. Then with probability at least $1 - e^{-c_2' m}$, the following holds:*

$$\sum_{i=1}^{|\mathcal{S}|} a_{[i],1}^2 \geq 0.99 |\mathcal{S}| \big[ 1 + \log(m/|\mathcal{S}|) \big] \tag{47}$$

*provided that $m \geq c_0 |\mathcal{S}| \geq c_1 n$ for some numerical constants $c_0$, $c_1$, $c_2' > 0$.*

Combining the results in Lemma 4 and (46) together, one further establishes that

$$\|\boldsymbol{Bx}\|^2 \geq |a_{[|\mathcal{S}|],1}|^{\gamma} \sum_{i=1}^{|\mathcal{S}|} a_{[i],1}^2 \geq |a_{[|\mathcal{S}|],1}|^{\gamma} \cdot 0.99 |\mathcal{S}| \big[ 1 + \log(m/|\mathcal{S}|) \big]. \tag{48}$$

The task remains to estimate the size of $|a_{[|\mathcal{S}|],1}|$, which we recall is the $|\mathcal{S}|$-th largest among the $m$ independent realizations $\{\psi_i = |a_{i,1}|\}_{1 \leq i \leq m}$. Taking $\gamma = -1$ in (45) gives the pdf of the half-normal distribution

$$p(t) = \sqrt{\frac{2}{\pi}} e^{-\frac{1}{2}t^2}, \quad t > 0 \tag{49}$$

whose corresponding cdf is

$$F(\tau) = \operatorname{erf}(\tau/\sqrt{2}). \tag{50}$$

Setting $F(\tau_{|\mathcal{S}|}) := 1 - |\mathcal{S}|/m$ or using the complementary cdf $|\mathcal{S}|/m := \operatorname{erfc}(\tau/\sqrt{2})$ based on the complementary error function gives rise to an estimate of the size of the $|\mathcal{S}|$-th largest [or equivalently, the $(m - |\mathcal{S}|)$-th smallest] entry in the $m$ realizations, namely

$$\tau_{|\mathcal{S}|} = \sqrt{2}\, \operatorname{erfc}^{-1}(|\mathcal{S}|/m) \tag{51}$$

where $\mathrm{erfc}^{-1}(\cdot)$ represents the inverse complementary error function. In the sequel, we show that the deviation of the $|\mathcal{S}|$-th largest realization $\psi_{|\mathcal{S}|}$ from its expected value $\tau_{|\mathcal{S}|}$ found above is bounded with high probability.

For random variable $\psi = |a|$ with $a$ obeying the standard Gaussian distribution, consider the event $\psi \leq \tau_{|\mathcal{S}|} - \delta$ for fixed constant $\delta > 0$. Define the indicator random variable $\chi = \mathbb{1}_{\{\psi \leq \tau_{|\mathcal{S}|} - \delta\}}$, whose expectation can be obtained by substituting $\tau = \tau_{|\mathcal{S}|} - \delta$ into the pdf in (50)

$$\mathbb{E}[\chi_i] = \mathrm{erf}(\tau_{|\mathcal{S}|} - \delta / \sqrt{2}). \tag{52}$$

Consider now the $m$ independent copies $\{\chi_i = \mathbb{1}_{\{\psi_i \leq \tau_{|\mathcal{S}|} - \delta\}}\}_{1 \leq i \leq m}$ of $\chi$, and the following holds

$$\mathbb{P}(\psi_{|\mathcal{S}|} \leq \tau_{|\mathcal{S}|} - \delta) = \mathbb{P}\Big( \sum_{i=1}^{m} \chi_i \leq m - |\mathcal{S}| \Big)$$

$$= \mathbb{P}\Big( \frac{1}{m} \sum_{i=1}^{m} \big( \chi_i - \mathbb{E}[\chi_i] \big) \leq 1 - \frac{|\mathcal{S}|}{m} - \mathbb{E}[\chi_i] \Big). \tag{53}$$

Clearly, random variables $\chi_i$ are bounded, so they are sub-gaussian [7]. For notational brevity, let $t := 1 - |\mathcal{S}|/m - \mathbb{E}[\chi_i] = 1 - |\mathcal{S}|/m - \mathrm{erf}(\tau_{|\mathcal{S}|} - \delta / \sqrt{2})$. Appealing to a large deviation inequality for sums of independent sub-gaussian random variables, one establishes that

$$\mathbb{P}(\psi_{|\mathcal{S}|} \leq \tau_{|\mathcal{S}|} - \delta) = \mathbb{P}\Big( \frac{1}{m} \sum_{i=1}^{m} \big( \chi_i - \mathbb{E}[\chi_i] \big) \leq 1 - \frac{|\mathcal{S}|}{m} - \mathbb{E}[\chi_i] \Big) \leq \mathrm{e}^{-c_5 m t^2} \tag{54}$$

where $c_5 > 0$ is some absolute constant. On the other hand, using the definition of the error function and properties of integration gives rise to

$$t = 1 - |\mathcal{S}|/m - \mathrm{erf}(\tau_{|\mathcal{S}|} - \delta / \sqrt{2}) = \frac{2}{\sqrt{\pi}} \int_{(\tau_{|\mathcal{S}|} - \delta)/\sqrt{2}}^{\tau_{|\mathcal{S}|}/\sqrt{2}} \mathrm{e}^{-s^2} \mathrm{d}s \geq \sqrt{\frac{2}{\pi}} \delta \mathrm{e}^{-\frac{\tau_{|\mathcal{S}|}^2}{2}} \geq \sqrt{\frac{2}{\pi}} \delta. \tag{55}$$

Taking the results in (54) and (55) together, one concludes that fixing any constant $\delta > 0$, the following holds with probability at least $1 - \mathrm{e}^{-c_2' m}$:

$$\psi_{|\mathcal{S}|} \geq \tau_{|\mathcal{S}|} - \delta \geq \sqrt{2} \, \mathrm{erfc}^{-1}(|\mathcal{S}|/m) - \delta$$

where the constant $c_2' := 2/\pi \cdot c_5 \delta^2$. Furthermore, choosing without loss of generality $\delta := 0.01\tau_{|\mathcal{S}|}$ above leads to $\psi_{|\mathcal{S}|} \geq 1.4 \, \mathrm{erfc}^{-1}(|\mathcal{S}|/m)$.

Substituting the last inequality into (48) and under our working assumption $|\mathcal{S}|/m \leq 0.25$, one readily obtains that

$$\|\boldsymbol{Bx}\|^2 \geq [1.4 \, \mathrm{erfc}^{-1}(|\mathcal{S}|/m)]^\gamma \cdot 0.99|\mathcal{S}|\big[1 + \log(m/|\mathcal{S}|)\big] \geq 0.99 \cdot 1.14^\gamma |\mathcal{S}|\big[1 + \log(m/|\mathcal{S}|)\big] \tag{56}$$

which holds with probability exceeding $1 - \mathrm{e}^{-c_2' m}$ for some absolute constant $c_2' > 0$, concluding the proof of Lemma 3.

## A.4 Proof of Proposition 3

To proceed, let us introduce the following events for all $1 \leq i \leq m$:

$$\mathcal{D}_i := \big\{ (\boldsymbol{a}_i^* \boldsymbol{x})(\boldsymbol{a}_i^* \boldsymbol{z}) < 0 \big\} \tag{57}$$

$$\mathcal{E}_i := \Big\{ \frac{|\boldsymbol{a}_i^* \boldsymbol{z}|}{|\boldsymbol{a}_i^* \boldsymbol{x}|} \geq \frac{1}{1 + \eta} \Big\} \tag{58}$$

for some fixed constant $\eta > 0$, in which the former corresponds to the gradients involving wrongly estimated signs, namely, $\frac{a_i^* z}{|a_i^* z|} \neq \frac{a_i^* x}{|a_i^* x|}$, and the second will be useful for deriving error bounds. Based on the definition of $\mathcal{D}_i$ and with $\mathbb{1}_{\mathcal{D}_i}$ denoting the indicator function of the event $\mathcal{D}_i$, we have

$$
\begin{aligned}
\langle \ell_{\mathrm{rw}}(z), h \rangle &= \frac{1}{m} \sum_{i=1}^{m} w_i \left( a_i^* z - |a_i^* x| \frac{a_i^* z}{|a_i^* z|} \right) (a_i^* h) \\
&= \frac{1}{m} \sum_{i=1}^{m} w_i \left( a_i^* h + a_i^* x - |a_i^* x| \frac{a_i^* z}{|a_i^* z|} \right) (a_i^* h) \\
&= \frac{1}{m} \sum_{i=1}^{m} w_i (a_i^* h)^2 + \frac{1}{m} \sum_{i=1}^{m} 2 w_i (a_i^* x)(a_i^* h) \mathbb{1}_{\mathcal{D}_i} \\
&\geq \frac{1}{m} \sum_{i=1}^{m} w_i (a_i^* h)^2 - \frac{1}{m} \sum_{i=1}^{m} 2 w_i |a_i^* x| |a_i^* h| \mathbb{1}_{\mathcal{D}_i}.
\end{aligned}
\tag{59}
$$

In the following, we will derive a lower bound for the term on the right hand side of (59). To be specific, a lower bound for the first term $\frac{1}{m} \sum_{i=1}^{m} w_i (a_i^* h)^2$ and an upper bound for the second term $\frac{1}{m} \sum_{i=1}^{m} 2 w_i |a_i^* x| |a_i^* h| \mathbb{1}_{\mathcal{D}_i}$ will be obtained, which occupies Lemmas 5 and 6, with their proofs postponed to Appendix A.5 and Appendix A.6, respectively.

**Lemma 5.** *Fix any $\eta, \beta > 0$. For any sufficiently small constant $\epsilon > 0$, the following holds with probability at least $1 - 2\mathrm{e}^{-c_5 \epsilon^2 m}$:*

$$
\frac{1}{m} \sum_{i=1}^{m} w_i (a_i^* h)^2 \geq \frac{1 - \zeta_1 - \epsilon}{1 + \beta(1 + \eta)} \|h\|^2
\tag{60}
$$

*with $w_i = \frac{1}{1 + \beta/(|a_i^* z|/|a_i^* x|)}$ for all $1 \leq i \leq m$, provided that $m/n > (c_6 \cdot \epsilon^{-2} \log \epsilon^{-1})$ for certain numerical constants $c_5, c_6 > 0$.*

Now we turn to the second term in (59). For ease of exposition, let us first introduce the following events

$$
\mathcal{B}_i := \left\{ |a_i^* x| < |a_i^* h| \leq (k+1)|a_i^* x| \right\}
\tag{61}
$$

$$
\mathcal{O}_i := \left\{ (k+1)|a_i^* x| < |a_i^* h| \right\}
\tag{62}
$$

for all $1 \le i \le m$ and some fixed constant $k > 0$. The second term can be bounded as follows

$$\frac{1}{m}\sum_{i=1}^{m} 2w_i|\boldsymbol{a}_i^*\boldsymbol{x}||\boldsymbol{a}_i^*\boldsymbol{h}|\mathbb{1}_{\mathcal{D}_i} \le \frac{1}{m}\sum_{i=1}^{m} w_i\left[(\boldsymbol{a}_i^*\boldsymbol{x})^2 + (\boldsymbol{a}_i^*\boldsymbol{h})^2\right]\mathbb{1}_{\{(\boldsymbol{a}_i^*\boldsymbol{z})(\boldsymbol{a}_i^*\boldsymbol{x})<0\}}$$

$$= \frac{1}{m}\sum_{i=1}^{m} w_i\left[(\boldsymbol{a}_i^*\boldsymbol{x})^2 + (\boldsymbol{a}_i^*\boldsymbol{h})^2\right]\mathbb{1}_{\{(\boldsymbol{a}_i^*\boldsymbol{h})(\boldsymbol{a}_i^*\boldsymbol{x})+(\boldsymbol{a}_i^*\boldsymbol{x})^2<0\}}$$

$$\le \frac{1}{m}\sum_{i=1}^{m} w_i\left[(\boldsymbol{a}_i^*\boldsymbol{x})^2 + (\boldsymbol{a}_i^*\boldsymbol{h})^2\right]\mathbb{1}_{\{|\boldsymbol{a}_i^*\boldsymbol{x}|<|\boldsymbol{a}_i^*\boldsymbol{h}|\}}$$

$$\le \frac{2}{m}\sum_{i=1}^{m} w_i(\boldsymbol{a}_i^*\boldsymbol{h})^2\mathbb{1}_{\{|\boldsymbol{a}_i^*\boldsymbol{x}|<|\boldsymbol{a}_i^*\boldsymbol{h}|\}}$$

$$= \frac{2}{m}\sum_{i=1}^{m} w_i(\boldsymbol{a}_i^*\boldsymbol{h})^2\mathbb{1}_{\{|\boldsymbol{a}_i^*\boldsymbol{x}|<|\boldsymbol{a}_i^*\boldsymbol{h}|\le(k+1)|\boldsymbol{a}_i^*\boldsymbol{x}|\}}$$

$$+ \frac{2}{m}\sum_{i=1}^{m} w_i(\boldsymbol{a}_i^*\boldsymbol{h})^2\mathbb{1}_{\{(k+1)|\boldsymbol{a}_i^*\boldsymbol{x}|<|\boldsymbol{a}_i^*\boldsymbol{h}|\}}$$

$$= \frac{2}{m}\sum_{i=1}^{m} w_i(\boldsymbol{a}_i^*\boldsymbol{h})^2\mathbb{1}_{\mathcal{B}_i} + \frac{2}{m}\sum_{i=1}^{m} w_i(\boldsymbol{a}_i^*\boldsymbol{h})^2\mathbb{1}_{\mathcal{O}_i} \qquad (63)$$

where the first equality is derived by substituting $\boldsymbol{z} = \boldsymbol{h} + \boldsymbol{x}$ according to the definition of $\boldsymbol{h}$, the second event suffices for $(\boldsymbol{a}_i^*\boldsymbol{h})(\boldsymbol{a}_i^*\boldsymbol{x}) + (\boldsymbol{a}_i^*\boldsymbol{x})^2 < 0$, and the second equality follows from writing the indicator function $\mathbb{1}_{\{|\boldsymbol{a}_i^*\boldsymbol{x}|<|\boldsymbol{a}_i^*\boldsymbol{h}|\}}$ as the summation of two indicator functions of two events $\mathbb{1}_{\{|\boldsymbol{a}_i^*\boldsymbol{x}|<|\boldsymbol{a}_i^*\boldsymbol{h}|\le(k+1)|\boldsymbol{a}_i^*\boldsymbol{x}|\}}$ and $\mathbb{1}_{\{|\boldsymbol{a}_i^*\boldsymbol{h}|>(k+1)|\boldsymbol{a}_i^*\boldsymbol{x}|\}}$.

The task so far remains to derive upper bounds for the two terms on the right side of (63), which leads to Lemma 6.

**Lemma 6.** *Fixing some $k > 0$, define $\zeta_2$ to be the maximum of $\mathbb{E}[w_i]$ in (72) for $\varrho = 0.01$ and $\nu = 0.1$, which depends only on $k$. For any $\epsilon > 0$, if $m/n > c_1'n\epsilon^{-2}\log\epsilon^{-1}$, the following hold simultaneously with probability at least $1 - c_3'\mathrm{e}^{-c_2'\epsilon^2 m}$:*

$$\frac{1}{m}\sum_{i=1}^{m} w_i(\boldsymbol{a}_i^*\boldsymbol{h})^2\mathbb{1}_{\mathcal{O}_i} \le (\zeta_2 + \epsilon)\|\boldsymbol{h}\|^2 \qquad (64)$$

*and*

$$\frac{1}{m}\sum_{i=1}^{m} w_i(\boldsymbol{a}_i^*\boldsymbol{h})^2\mathbb{1}_{\mathcal{B}_i} \le \frac{0.1271 - \zeta_2 + \epsilon}{1 + \beta/k}\|\boldsymbol{h}\|^2 \qquad (65)$$

*for all $\boldsymbol{h} \in \mathbb{R}^n$ obeying $\|\boldsymbol{h}\|/\|\boldsymbol{x}\| \le 1/10$, where $c_1'$, $c_2'$, $c_3' > 0$ are some absolute constants.*

For a few $k$ values, the corresponding $\xi_2$ values are listed as follows: When $k \in \{1, 2, 3, 4, 5, 9, 10\}$, then $\xi_2$ takes values in $\{0.0639, 0.0426, 0.0320, 0.0256, 0.0213,$
$0.0128, 0.0116\}$ accordingly.

Taking the results in (60), (63), and (64)-(65) established in Lemmas 5 and 6 back into (59), we

conclude that

$$\langle \ell_{\text{rw}}(\boldsymbol{z}), \boldsymbol{h} \rangle \geq \frac{1}{m} \sum_{i=1}^{m} w_i (\boldsymbol{a}_i^* \boldsymbol{h})^2 \mathbb{1}_{\mathcal{E}_i} - \frac{1}{m} \sum_{i=1}^{m} 2 w_i |\boldsymbol{a}_i^* \boldsymbol{x}| |\boldsymbol{a}_i^* \boldsymbol{h}| \mathbb{1}_{\mathcal{D}_i}$$

$$\geq \frac{1 - \zeta_1 - \epsilon}{1 + \beta(1 + \eta)} \|\boldsymbol{h}\|^2 - 2(\zeta_2 + \epsilon) \|\boldsymbol{h}\|^2 - \frac{2(0.1271 - \zeta_2 + \epsilon)}{1 + \beta/k} \|\boldsymbol{h}\|^2$$

$$= \left[ \frac{1 - \zeta_1 - \epsilon}{1 + \beta(1 + \eta)} - 2(\zeta_2 + \epsilon) - \frac{2(0.1271 - \zeta_2 + \epsilon)}{1 + \beta/k} \right] \|\boldsymbol{h}\|^2 \qquad (66)$$

which will be rendered positive, if $\beta > 0$ is small enough, and parameters $\eta > 0$, $k > 0$ are suitably chosen.

## A.5   Proof of Lemma 5

Plugging in the weighting parameters $w_i = \frac{1}{1 + \beta/(|\boldsymbol{a}_i^* \boldsymbol{z}|/|\boldsymbol{a}_i^* \boldsymbol{x}|)}$ and based on the definition of $\mathcal{E}_i$, the first term in (59) can be lower bounded as follows

$$\frac{1}{m} \sum_{i=1}^{m} w_i (\boldsymbol{a}_i^* \boldsymbol{h})^2 \geq \frac{1}{m} \sum_{i=1}^{m} \frac{1}{1 + \beta/(|\boldsymbol{a}_i^* \boldsymbol{z}|/|\boldsymbol{a}_i^* \boldsymbol{x}|)} (\boldsymbol{a}_i^* \boldsymbol{h})^2 \mathbb{1}_{\mathcal{E}_i} \qquad (67)$$

$$\geq \frac{1}{m} \sum_{i=1}^{m} \frac{1}{1 + \beta(1 + \eta)} (\boldsymbol{a}_i^* \boldsymbol{h})^2 \mathbb{1}_{\left\{ \frac{|\boldsymbol{a}_i^* \boldsymbol{z}|}{|\boldsymbol{a}_i^* \boldsymbol{x}|} \geq \frac{1}{1+\eta} \right\}}$$

$$= \frac{1}{1 + \beta(1 + \eta)} \cdot \frac{1}{m} \sum_{i=1}^{m} (\boldsymbol{a}_i^* \boldsymbol{h})^2 \mathbb{1}_{\mathcal{E}_i} \qquad (68)$$

where the first inequality arises from excluding some nonnegative terms from the left hand side, and the second one replaced the ratio $|\boldsymbol{a}_i^* \boldsymbol{z}|/|\boldsymbol{a}_i^* \boldsymbol{x}|$ in the weights by its lower bound $1/1+\eta$ because the weights are monotonically increasing functions of the ratios $|\boldsymbol{a}_i^* \boldsymbol{z}|/|\boldsymbol{a}_i^* \boldsymbol{x}|$. Using the result in Lemma 7, the last term in (68) can be further bounded by

$$\frac{1}{m} \sum_{i=1}^{m} w_i (\boldsymbol{a}_i^* \boldsymbol{h})^2 \geq \frac{1}{1 + \beta(1 + \eta)} \cdot \frac{1}{m} \sum_{i=1}^{m} (\boldsymbol{a}_i^* \boldsymbol{h})^2 \mathbb{1}_{\mathcal{E}_i} \geq \frac{1 - \zeta_1 - \epsilon}{1 + \beta(1 + \eta)} \|\boldsymbol{h}\|^2 \qquad (69)$$

for any fixed sufficiently small constant $\epsilon > 0$, which holds with probability at least $1 - 2\mathrm{e}^{-c_5 \epsilon^2 m}$, if $m > (c_6 \cdot \epsilon^{-2} \log \epsilon^{-1}) n$.

## A.6   Proof of Lemma 6

The proof is adapted from that of [11, Lemma 3]. We first prove the bound (64) for any fixed $\boldsymbol{h}$ obeying $\|\boldsymbol{h}\| \leq \|\boldsymbol{x}\|/10$, and subsequently develop a uniform bound at the end of this section. The bound (65) can be derived directly from subtracting the bound in (64) with $k$ from that bound with $k = 0$, followed by an application of the Bernstein-type sub-exponential tail bound [7]. Hence, we only discuss the first bound (64). Because of the discontinuity hence non-Lipschitz of the indicator functions, let us approximate them by a sequence of auxiliary Lipschitz functions. Specifically, with some constant $\varrho > 0$, define for all $1 \leq i \leq m$ the ensuing continuous functions

$$\chi_i(s) := \begin{cases} s, & s > (1+k)^2 (\boldsymbol{a}_i^* \boldsymbol{x})^2 \\ \frac{1}{\varrho}[s - (k+1)^2 (\boldsymbol{a}_i^* \boldsymbol{x})^2] \\ \quad + (k+1)^2 (\boldsymbol{a}_i^* \boldsymbol{x})^2, & (1 - \varrho)(k+1)^2 (\boldsymbol{a}_i^* \boldsymbol{x})^2 \leq s \leq (k+1)^2 (\boldsymbol{a}_i^* \boldsymbol{x})^2 \\ 0, & \text{otherwise.} \end{cases} \qquad (70)$$

Clearly, all $\chi_i(\mathrm{s})$'s are random Lipschitz functions with constant $^1\!/\!_\varrho$. Furthermore, it is easy to verify that

$$|\boldsymbol{a}_i^*\boldsymbol{h}|^2 \mathbb{1}_{\{(k+1)|\boldsymbol{a}_i^*\boldsymbol{x}|<|\boldsymbol{a}_i^*\boldsymbol{h}|\}} \leq \chi_i(|\boldsymbol{a}_i^*\boldsymbol{h}|^2) \leq |\boldsymbol{a}_i^*\boldsymbol{h}|^2 \mathbb{1}_{\{\sqrt{1-\varrho}(k+1)|\boldsymbol{a}_i^*\boldsymbol{x}|<|\boldsymbol{a}_i^*\boldsymbol{h}|\}}. \tag{71}$$

Given that the second term involves the addition event $\mathcal{G}_i$ in (58), define $w_i := \dfrac{|\boldsymbol{a}_i^*\boldsymbol{h}|^2}{\|\boldsymbol{h}\|^2}\mathbb{1}_{\{\sqrt{1-\varrho}(k+1)|\boldsymbol{a}_i^*\boldsymbol{x}|<|\boldsymbol{a}_i^*\boldsymbol{h}|\}}$ for $1 \leq i \leq m$, and also $\nu := \dfrac{\|\boldsymbol{h}\|}{\|\boldsymbol{x}\|}$ for notational convenience. If $f(\tau_1, \tau_2)$ denotes the density of two joint Gaussian random variables with correlation constant $\rho = \dfrac{\boldsymbol{h}^*\boldsymbol{x}}{\|\boldsymbol{h}\|\|\boldsymbol{x}\|} \in (-1, 1)$, then the expectation of $w_i$ can be obtained based on the conditional expectation

$$\mathbb{E}[w_i] = \int_{-\infty}^{\infty} \mathbb{E}[w_i | \boldsymbol{a}_i^*\boldsymbol{x} = \tau_1\|\boldsymbol{x}\|, \boldsymbol{a}_i^*\boldsymbol{h} = \tau_1\|\boldsymbol{h}\|]f(\tau_1, \tau_2)\mathrm{d}\tau_1\mathrm{d}\tau_2$$

$$= \int_{-\infty}^{\infty} \int_{-\infty}^{\infty} \tau_2^2 \mathbb{1}_{\{\sqrt{1-\varrho}(k+1)|\tau_1|<|\tau_2|\nu\}} f(\tau_1, \tau_2)\mathrm{d}\tau_1\mathrm{d}\tau_2$$

$$= \frac{1}{\sqrt{2\pi}} \int_0^{\infty} \tau_2^2 \exp(-\tau_2^2/2)\left[\mathrm{erf}\left(\frac{(\nu/[\sqrt{1-\varrho}(k+1)]-\rho)\tau_2}{\sqrt{2(1-\rho^2)}}\right)\right.$$

$$\left. + \mathrm{erf}\left(\frac{(\nu/[\sqrt{1-\varrho}(k+1)]+\rho)\tau_2}{\sqrt{2(1-\rho^2)}}\right)\right]\mathrm{d}\tau_2 \tag{72}$$

$$:= \zeta_2. \tag{73}$$

It is not difficult to see that $\mathbb{E}[w_i] = 0$ for $\rho = \pm 1$, and $\mathbb{E}[w_i]$ is continuous over $\rho \in (-1, 1)$ due to the integration property of continuous functions over a continuous interval. Although the last term in (72) can not be expressed in closed-form, it can be evaluated numerically. Note first that for fixed parameters $\varrho > 0$ and $\nu \leq 0.1$, the integration above is monotonically decreasing in $k \geq 0$, and achieves the maximum at $k = 0$. For parameter values $k = 5$, $\nu = 0.1$ and $\varrho = 0.01$, Fig. 1 plots $\mathbb{E}[w_i]$ as a function of $\rho$, whose maximum $\zeta_2 = 0.0213$. is achieved at $\rho = 0$. Further from the integration in (72), for fixed $k \geq 0$, $\mathbb{E}[w_i]$ is a monotonically increasing function of both $\nu$ and $\varrho$, it is therefore safe to conclude that for all $0 < \nu \leq 0.1$, and $\varrho = 0.01$, we have

$$\mathbb{E}[w_i] \leq \zeta_2 = 0.0213. \tag{74}$$

Hence, it is safe to conclude that $\mathbb{E}[\chi_i(|\boldsymbol{a}_i^*\boldsymbol{h}|^2)] \leq 0.0213\|\boldsymbol{h}\|^2$ for $\nu < 0.1$, $\varrho = 0.01$, and $k = 5$. Since $[\chi_i(|\boldsymbol{a}_i^*\boldsymbol{h}|^2$'s are sub-exponential with sub-exponential norm of the order $\mathcal{O}(\|\boldsymbol{h}\|^2)$, Bernstein-type sub-exponential tail bound [7] confirms that

$$\mathbb{P}\left(\frac{1}{m}\sum_{i=1}^m \frac{\chi_i(|\boldsymbol{a}_i^*\boldsymbol{h}|^2)}{\|\boldsymbol{h}\|^2} > (\zeta_2 + \epsilon)\right) < \mathrm{e}^{-c_7 m\epsilon^2} \tag{75}$$

for some numerical constant $\epsilon > 0$, provided that $\|\boldsymbol{h}\| \leq \|\boldsymbol{x}\|/_{10}$. Finally, due to the fact that $w_i \leq 1$ for all $1 \leq i \leq m$, the following holds

$$\frac{1}{m}\sum_{i=1}^m w_i\chi_i(|\boldsymbol{a}_i^*\boldsymbol{h}|^2) < (\zeta_2 + \epsilon)\|\boldsymbol{h}\|^2 \tag{76}$$

with probability at least $1 - \mathrm{e}^{-c_7 m\epsilon^2}$.

We have proved the bound in (64) for a fixed vector $\boldsymbol{h}$, and the uniform bound for all vectors $\boldsymbol{h}$ obeying $\|\boldsymbol{h}\| \leq \|\boldsymbol{x}\|/_{10}$ can be obtained by similar arguments in the proof [11, Lemma 9] with only minor changes in the constants.

Figure 1: The expectation $\mathbb{E}[w_i]$ as a function of $\rho$ over $[-1, 1]$.

Regarding the second bound (65), it is easy to see that

$$\frac{1}{m}\sum_{i=1}^{m}|\boldsymbol{a}_i^*\boldsymbol{h}|^2\mathbb{1}_{\{|\boldsymbol{a}_i^*\boldsymbol{x}|<|\boldsymbol{a}_i^*\boldsymbol{h}|\leq(k+1)|\boldsymbol{a}_i^*\boldsymbol{x}|\}} = \frac{1}{m}\sum_{i=1}^{m}|\boldsymbol{a}_i^*\boldsymbol{h}|^2\mathbb{1}_{\{|\boldsymbol{a}_i^*\boldsymbol{x}|<|\boldsymbol{a}_i^*\boldsymbol{h}|\}}$$

$$-\frac{1}{m}\sum_{i=1}^{m}|\boldsymbol{a}_i^*\boldsymbol{h}|^2\mathbb{1}_{\{(k+1)|\boldsymbol{a}_i^*\boldsymbol{x}|<|\boldsymbol{a}_i^*\boldsymbol{h}|\}}$$

$$\leq (0.1271 - \zeta_2 + \epsilon)\|\boldsymbol{h}\|^2 \qquad (77)$$

where the last inequality follows from subtracting the bound in (64) of $k$ from that corresponding to $k = 0$. To account for the weights $w_i = \frac{1}{1+\beta/(|\boldsymbol{a}_i^*\boldsymbol{z}|/|\boldsymbol{a}_i^*\boldsymbol{x}|)}$, first notice that $\boldsymbol{a}_i^*\boldsymbol{h} = \boldsymbol{a}_i^*\boldsymbol{z} - \boldsymbol{a}_i^*\boldsymbol{x}$, and that our second bound works with $(\boldsymbol{a}_i^*\boldsymbol{z})(\boldsymbol{a}_i^*\boldsymbol{x}) < 0$ in (59), hence $\frac{|\boldsymbol{a}_i^*\boldsymbol{z}|}{|\boldsymbol{a}_i^*\boldsymbol{x}|} \leq \frac{|\boldsymbol{a}_i^*\boldsymbol{h}|}{|\boldsymbol{a}_i^*\boldsymbol{x}|} - 1$. Recall that the second bound (65) assumes the event $\{|\boldsymbol{a}_i^*\boldsymbol{x}| < |\boldsymbol{a}_i^*\boldsymbol{h}| \leq (k+1)|\boldsymbol{a}_i^*\boldsymbol{x}|\}$, implying $\frac{|\boldsymbol{a}_i^*\boldsymbol{z}|}{|\boldsymbol{a}_i^*\boldsymbol{x}|} \leq \frac{|\boldsymbol{a}_i^*\boldsymbol{h}|}{|\boldsymbol{a}_i^*\boldsymbol{x}|} - 1 \leq k$. Further, because $w_i$ is monotonically increasing in $\frac{|\boldsymbol{a}_i^*\boldsymbol{z}|}{|\boldsymbol{a}_i^*\boldsymbol{x}|}$, then $w_i \leq \frac{1}{1+\beta/k}$. Taking this result back to (77) yields

$$\frac{1}{m}\sum_{i=1}^{m}w_i|\boldsymbol{a}_i^*\boldsymbol{h}|^2\mathbb{1}_{\{|\boldsymbol{a}_i^*\boldsymbol{x}|<|\boldsymbol{a}_i^*\boldsymbol{h}|\leq(k+1)|\boldsymbol{a}_i^*\boldsymbol{x}|\}} \leq \frac{0.1271 - \zeta_2 + \epsilon}{1 + \beta/k}\|\boldsymbol{h}\|^2 \qquad (78)$$

which proves the second bound in (65).

**Lemma 7.** *([10, Lemma 5]) Fix $\eta \geq 1/2$ and $\rho \leq 1/10$, and let $\mathcal{E}_i$ be defined in (57). For independent random variables $Y \sim \mathcal{N}(0, 1)$ and $Z \sim \mathcal{N}(0, 1)$, define*

$$\zeta_1 := 1 - \min\left\{\mathbb{E}\left[\mathbb{1}_{\left\{\left|\frac{1-\rho}{\rho}+\frac{Y}{Z}\right|\geq\frac{\sqrt{1.01}}{\rho(1+\eta)}\right\}}\right], \mathbb{E}\left[Z^2\mathbb{1}_{\left\{\left|\frac{1-\rho}{\rho}+\frac{Y}{Z}\right|\geq\frac{\sqrt{1.01}}{\rho(1+\eta)}\right\}}\right]\right\}. \qquad (79)$$

*Fixing any $\epsilon > 0$ and for any $\boldsymbol{h}$ satisfying $\|\boldsymbol{h}\|/\|\boldsymbol{x}\| \leq \rho$, the next holds with probability $1 - 2\mathrm{e}^{-c_5\epsilon^2 m}$:*

$$\frac{1}{m}\sum_{i=1}^{m}(\boldsymbol{a}_i^*\boldsymbol{h})^2 \mathbb{1}_{\mathcal{E}_i} \geq (1 - \zeta_1 - \epsilon)\|\boldsymbol{h}\|^2 \tag{80}$$

*provided that $m > (c_6 \cdot \epsilon^{-2}\log\epsilon^{-1})n$ for some universal constants $c_5$, $c_6 > 0$.*

To have an estimate of the size of $\xi_1$ in (79), if $\gamma = 0.7$ and $\rho = 1/10$, we have $\mathbb{E}\left[\mathbb{1}_{\left\{\left|\frac{1-\rho}{\rho} + \frac{Y}{Z}\right| \geq \frac{\sqrt{1.01}}{\rho(1+\gamma)}\right\}}\right] \approx$ 0.9216, and $\mathbb{E}\left[Z^2\mathbb{1}_{\left\{\left|\frac{1-\rho}{\rho} + \frac{Y}{Z}\right| \geq \frac{\sqrt{1.01}}{\rho(1+\gamma)}\right\}}\right] \approx 0.9908$, hence leading to $\zeta_1 \approx 0.0784$.