[Reviews · NeurIPS 2017]

Reviewer 1



The paper proposes a novel algorithm for phase retrieval that first finds a good initialization with maximum correlation and then refines the solution with gradient-like updates. The algorithm is guaranteed with high probability to converge to the true solution under noiseless setting and with bounded error under noise setting for most Gaussian sensing matrices. A highlight of the contribution is that the guarantee still holds even for small number of measurements near the information-theoretic limit. Experiments also demonstrate its practical success compared with existing methods. The paper is very well written. It clearly outlines the problem, explains the difference and contributions compared with existing works, and presents good intuition why the algorithm works well. The contributions from the two stages, both initialization and local refinement, seem to be novel enough to allow better statistical accuracies with few number of measurements near information-theoretic limit and lead better empirical results.

Reviewer 2



Update: I've increased my rating from 6-7 because the authors did a good job addressing my concerns, and the distinction between theoretical and empirical claims seems a bit better upon a fresh read. One more comment: Given the empirical nature of the experiments, I think the authors should have compared to state-of-the-art phase retrieval methods that are known to out-perform WF methods (like Feinup's HIO method or Gerchberg–Saxton). It's likely that such methods would perform at the information theoretic limit as well. ============================================= This paper presents a (somewhat) new method for solving phase retrieval problems that builds off of other phase retrieval schemes. The method has 2 phases: 1) Construct an initial guess of the solution using a spectral initialization scheme 2) Refine the guess using a gradient descent method. The new method is interesting, and empirically the method performs well. My reservations about the paper stem from the fact that the claims it makes are dramatically over-sold (and possibly untrue, depending on interpretation), and gloss over existing contributions that others have made in this field. This is a good paper with interesting results. I rate it as "marginal accept" instead of "accept" because the presentation is at times misleading and ungenerous to other authors (who's work they build upon extensively). 1) The authors appear to be claiming to be the first method to solve "almost" all quadratic equations. However, several other methods exists with similar (identical?) theoretical strength. The authors don't seem to acknowledge this fact until line 266. 2) In the abstract and intro, the authors make no distinction between empirical and theoretical guarantees. This paper provides no theoretical guarantees of achieving the optimal number of measurements. Any observations of this fact are purely empirical. Empirical claims are fine with me, but the authors should be up-front about the fact that their results are only observed in experiments (the first time I read this paper the abstract and intro led me to believe you have a *proof* of achieving the lower bound, which you do not). 3) Speaking of experiments, how did the authors choose the truncation parameters for the TWF comparison? Did they tune this parameter for optimal performance? The parameter should be reported to make experiments repeatable. 4) You claim that recovery takes place in time proportional to MxN. I'm not sure this is true. The convex optimization step can definitely be done in NxM time, but what about the (very important) initialization step? I don't think you can make this claim unless you have a constant upper-bound on the amount of work needed to obtain the leading eigenvector (for example a constant bound on the number of power iterations that does not depend on N or M). The authors should either justify this claim or remove it. Or maybe the claim is only based on empirical experiments (in which case this should be made clear)? Finally, the authors should make clear that truncated WF flow methods have the same runtime bounds as this method (although I think the truncated WF paper avoids making any claims of linear overall runtime because of the issues mentioned above with the initializer). Also, you might want to cite the following (optional - I leave this up to the authors): - "Phase Transitions of Spectral Initialization for High-Dimensional Nonconvex Estimation" This paper does a rigorous analysis of spectral initiation methods, and proves tight bounds that don't require constants. This is relevant because the empirical results you got in Fig1 are predicted by this theory. - On line 66: You mention that convex methods require large storage and computation. The recent papers "Phase Retrieval Meets Statistical Learning Theory: A Flexible Convex Relaxation" and "PhaseMax: Convex Phase Retrieval via Basis Pursuit" use convexity without the extra storage. I doubt, though, that these methods can be performed with provably linear time complexity. - "Phase Transitions of Spectral Initialization for High-Dimensional Nonconvex Estimation" . This is an alternative formulation for truncated WF that has gotten some attention. I think it's worth mentioning since this is in the same class of algorithms as the proposed method, and is very closely related.